# A Discrete Schwarzian Derivative via Circle Packing

Kenneth Stephenson

Department of Mathematics, University of Tennessee, Knoxville, TN 37996, USA; kstephe2@utk.edu

**Abstract**

There exists an extensive and fairly comprehensive discrete analytic function theory which is based on circle packing. This paper introduces a faithful discrete analogue of the classical Schwarzian derivative to this theory and develops its basic properties. The motivation comes from the current lack of circle packing algorithms in spherical geometry, and the discrete Schwarzian derivative may provide for new approaches. A companion localized notion called an intrinsic schwarzian is also investigated. The main concrete results of the paper are limited to circle packing flowers. A parameterization by intrinsic schwarzians is established, providing an essential packing criterion for flowers. The paper closes with the study of special classes of flowers that occur in the circle packing literature. As usual in circle packing, there are pleasant surprises at nearly every turn, so those not interested in circle packing theory may still enjoy the new and elementary geometry seen in these flowers.

**Keywords:** circle packing; Schwarzian derivative; discrete analytic functions; Möbius transformations

**MSC:** 30G25; 52C26

## 1. Introduction

Classical complex analysis, and conformal geometry in general, have long benefited from a fundamental Möbius invariant known as the Schwarzian derivative. Recent decades have seen the emergence of a comprehensive discrete analytic function theory and associated discrete conformal geometry based on circle packing. This discrete theory displays deep and intimate connections to conformal geometry, so it is natural to ask whether it, too, could benefit from such an invariant. This paper establishes definitions for a discrete Schwarzian derivative and verifies fundamental properties that are largely faithful to the classical version. It also introduces a local Möbius invariant, an intrinsic schwarzian, and begins to lay out how these invariants might provide important tools in advancing the theory of circle packing.

Möbius or projective invariance is exemplified by quantities which remain unchanged after application of Möbius transformations. While the Riemann sphere $\mathbb{P}$ is the native habitat for Möbius actions, it is also far and away the most challenging for circle packing. Indeed, with a few exceptions, circle packings on $\mathbb{P}$ have been merely stereographic projections of the packings developed in the Euclidean or hyperbolic setting. These spherical difficulties account for perhaps the most glaring gap in discrete analytic function theory, namely our inability to create and manipulate discrete rational functions.

The circle packing community has exhausted most approaches to working in spherical geometry, with precious little to show for it. Perhaps discrete Schwarzian derivatives can

provide the fresh perspective needed to move forward. The reader should not expect miracles, however. Although we do establish robust definitions and basic properties for a discrete Schwarzian derivative, taking our lead from pioneering work by Gerald Orick, and although we take the opening steps, there are no breakthrough theoretical tools here. On the other hand, in the experimental world available via circle packing, the discrete Schwarzian derivative and the associated intrinsic schwarzian open wholly new vistas for investigation. Concrete results here mostly deal with the fundamental unit within every circle packing, namely the circle packing "flower". As invariably happens in circle packing, both beautiful visualizations and beautiful formulas pop up around every corner. Whether or not the reader is involved in circle packing theory, there is much to appreciate in the surprising and pleasing elementary geometry we encounter. And we can always be alert for that breakout tool.

Here is a brief overview of the paper: We first provide necessary (but brief) background on circle packing, on the associated discrete analytic functions, on geometry and Möbius transformations, and on the central role that experiments play in this topic. In Section 2, we review the classical Schwarzian derivative and define a discrete version for mappings between circle packings. Moving beyond that direct analog, we extract a local version, an *intrinsic schwarzian,* attached to individual packings. A principal goal—a distant goal—is methods for recognizing, creating, and ultimately manipulating (intrinsic) schwarzians for packings. These schwarzians form edge labels analogous to the vertex (i.e., radius) labels which dominate the theory, but which largely fail on the sphere. Section 3 illustrates the as-of-yet-unfulfilled potential for schwarzians as a mechanism for laying out circle packings. The struggle to work with discrete meromorphic functions is our main motivation, but results could also apply to circle packings on projective surfaces.

We switch in Section 5 to the paper's modest results from our opening skirmishes with schwarzians; namely, describing the schwarzians for flowers, the elemental circle packings. An $n$-flower consists of a central circle surrounded by a chain of $n$ tangent "petal" circles. A flower is *un-branched* if the petals wrap once around the center and *branched* if they wrap two or more times. It is *univalent* if un-branched and the petals have mutually disjoint interiors. Using a mechanical layout process and the computations detailed in Appendix A.1 we work our way through the early cases $n = 3, 4, 5$, and 6 to general flowers. We reach characterizations of un-branched (Theorem 2) and univalent (Theorem 3) flowers and criteria for branching.

We conclude the paper in Section 6 by applying what we have learned to several special classes of flowers. These cases will contribute only marginally to the larger campaign, but they raise our spirits with beautiful geometric, visual, and arithmetic features. And although much remains to be done, in the author's view, the results for flowers alone are worth the effort.

## 2. Background

### 2.1. On Circle Packing

A *circle packing* is a configuration of circles satisfying a prescribed pattern of tangencies. Circle packings and their connections to conformal geometry were introduced by William Thurston in 1985 [1]. Circle packings exist in great profusion in Euclidean, hyperbolic, and spherical geometry, and more recently on surfaces with affine and projective structures [2,3]. The principal reference for this paper is [4].

The fundamental machinery is quite straightforward: The pattern of tangencies for a circle packing $P$ is encoded in an abstract (simplicial) complex $K$, a triangulation of a topological surface. There is a circle $C_v \in P$ associated with each vertex $v$ of $K$ and each edge $\langle v, w \rangle$ of $K$ indicates a required tangency between circles $C_v$ and $C_w$. Note that every

"circle" is associated with an interior, forming a topological disc. Two circles are (externally) tangent if they intersect in a single point and their interiors are mutually disjoint. Often, the key data associated with a packing is a radius label $R$, which contains a radius $R(v)$ for the circle associate with vertex $v \in K$.

Some basic terminology will be useful in the sequel: The circles of a packing $P$ occur in mutually tangent triples $\{C_v, C_w, C_u\}$. The geodesics connecting the three centers pass through the three tangency points and form a geometric face. This is a geometric triangle associated with the abstract *face* $\{v, w, u\}$ of $K$. The surface formed by the geometric faces is called the *carrier* of $P$. The packing $P$ is *univalent* if its circles have mutually disjoint interiors.

The packing $P$ can also be viewed as a collection of interconnected flowers: a *flower* consists of a *central* circle $C_v$ and the chain of successively tangent *petal* circles, $\{C_{v_0}, \cdots, C_{v_{n-1}}\}$, all tangent to $C_v$. A flower is *closed* if $C_{v_{n-1}}$ is tangent to $C_{v_0}$, in which case $v$ is an *interior* vertex of $K$, whereas a flower is *open* if and only if $v$ is a *boundary* vertex. (To avoid pathologies, we require of $K$ that every boundary vertex has at least one interior neighbor.) There are three classes of closed flowers: A *univalent* flower is one whose petals have mutually disjoint interiors. An *un-branched* flower is one whose petals wrap once around the center, possibly with overlaps between non-contiguous petals. Finally, a *branched* flower is one whose petals wrap more than once about the center and its *degree d* is the number of times it wraps.

A circle packing $P$ is *univalent* if its circles have mutually disjoint interiors. It is necessary (but not sufficient) that the flowers for interior vertices are univalent. If an interior circle $C_v$ has a branched flower, then we say that $P$ has a branch point at $v$.

The surprising richness of the topic is seen in the foundational existence and uniqueness result; namely the Koebe–Andreev–Thurston (KAT) Theorem, which states that, for any triangulation $K$ of a topological sphere, there exists an associated univalent circle packing $\mathcal{P}_K$ of the Riemann sphere $\mathbb{P}$, and that $\mathcal{P}_K$ is unique up to Möbius transformations (and inversions) of $\mathbb{P}$. Thurston also proposed in [1] a clever algorithm for actually computing such packings, allowing us today to treat circle packing as a verb: to "circle pack" a complex $K$ is to create and manipulate the associated circle packings.

*2.2. On Discrete Analytic Functions*

Intriguing as circle packings were in their own right, it was a conjecture in Thurston's talk that really fired up the topic. An example in Figure 1 will set the stage. Let $P$ be a univalent circle packing filling a simply connected region $\Omega$ of the plane, as on the right in the figure, and let $K$ be the underlying complex. Thurston proved using KAT that there exists a univalent circle packing for $K$ in the unit disc $\mathbb{D}$ whose boundary circles are all horocycles, as on the left in the figure. With two circle packings for the same complex, one may define $F : \mathcal{P}_K \longrightarrow P$ by identifying the corresponding circles. Essentially a mapping from the unit disc to $\Omega$, $F$ is roughly analogous to the classical Riemann Mapping.

It is the conjecture Thurston made about such discrete conformal mappings that kicked off the nearly 40 years of development efforts in circle packing. He suggested that, if one were to refine this construction—used circle packings $P$ with ever more and smaller circles—that the resulting circle packing maps $f$ would converge uniformly on a compact subset of $\mathbb{D}$ to the classical Riemann mapping from $\mathbb{D}$ onto $\Omega$. Shortly thereafter, this was proven by B. Rodin and D. Sullivan [5] in the case of hexagonal circle packings. This result has subsequently been expanded to nearly full generality by many authors; see [4] for the story.

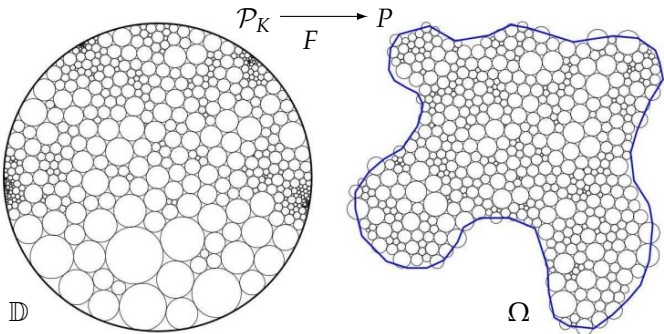

**Figure 1.** Example of a discrete Riemann mapping.

The packing $\mathcal{P}_K$ has come to be called the *maximal* packing for *K*. The range of settings has vastly expanded, so the existence and uniqueness of appropriate maximal packings has been proven for essentially any complex *K*, whether finite, infinite, multiply connected, with or without boundary. A complex *K* may, of course, support other circle packings. When two packings, say *Q* and *P*, share *K*, then the mapping $F : Q \longrightarrow P$ is known as a *discrete analytic function*. As mentioned earlier, the mapping in Figure 1 is a *discrete Riemann mapping*. However, there are discrete analogues available for nearly all types of analytic functions, from entire functions to universal covering maps, and even branched functions. Moreover, the convergence of the discrete maps under refinement to their classical counterparts is established in nearly every circumstance. One can rightly think of this as "quantum" complex analysis—a discrete theory which not only mimics the classical, but also converges to it under refinement.

Missing, however, in the pantheon of discrete analytic functions is the potentially rich family of discrete meromorphic functions. There is no mystery in the appropriate definition on the sphere: If *K* triangulates a sphere and *P* is a circle packing for *K* on $\mathbb{P}$, then the map $F : \mathcal{P}_K \longrightarrow P$ would be a *discrete meromorphic function*. Figure 2a is a non-trivial example that we will return to in the sequel. Discrete meromorphic functions can appear more generally as well: Figure 2b represents a discrete meromorphic function mapping a torus to the sphere.

Both these examples owe their existence to combinatorial symmetries. The first, a discrete analog of the classical meromorphic function $z^3(3z^5 - 1)/(z^5 + 3)$, exploits dodecahedral symmetry and the special geometry of *Schwarz triangles* in $\mathbb{P}$. There are 12 branched circles, and each has 5 petal circles wrapping twice around it; an isolated flower will be shown later when we revisit this example.

The sphere packing of Figure 2b, developed jointly with Edward Crane, is a discrete version of a Weierstrass $\wp$ function, mapping a torus to a 2-sheeted covering of $\mathbb{P}$ with four simple branch points (the colored circles). There is a special symmetry built into its complex *K* and the choice of branch vertices, though we have yet to understand fully why this symmetry ensures coherent circle packing in $\mathbb{P}$.

Absent special symmetries, creating such non-univalent packings is out of reach, with inherent difficulties in spherical geometry compounded by the need for branch points. Methods for constructing packings of $\mathbb{P}$, and more generally, packings on Riemann surfaces with projective structures, are the principal motivation for this work.

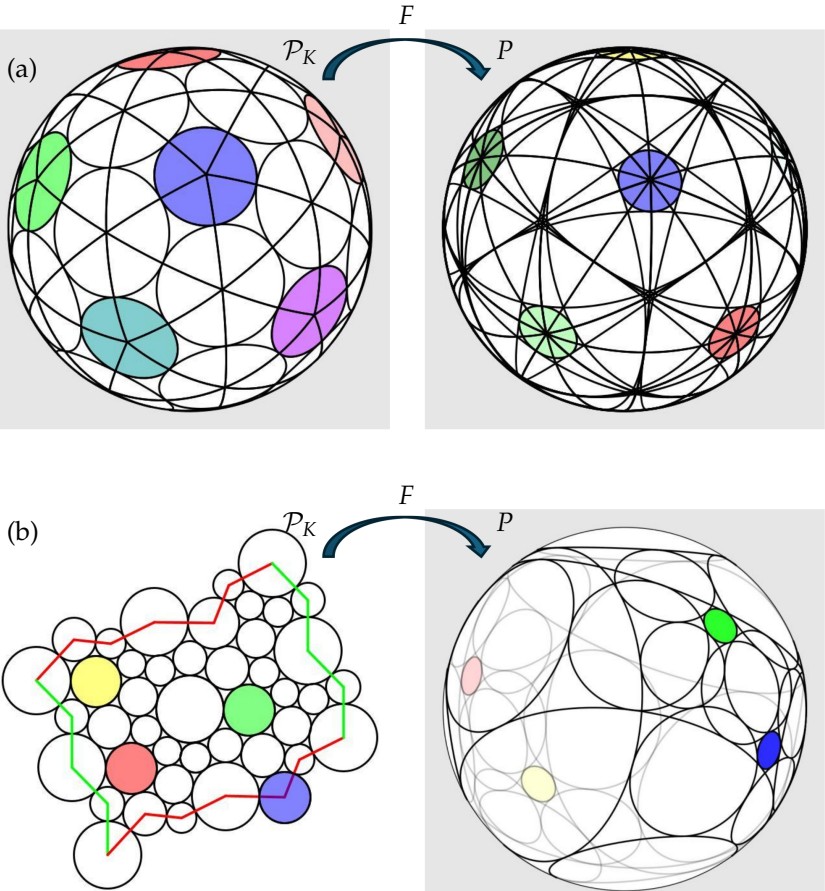

**Figure 2.** Examples of discrete meromorphic functions: (**a**) a discrete rational function; (**b**) a discrete Weierstrass $\wp$ function.

### 2.3. On Möbius Transformations

Spherical geometry refers here to the geometry of the *Riemann sphere* $\mathbb{P}$, also known as the *complex projective line*. We model $P$ on the unit sphere centered at the origin in $\mathbb{R}^3$ and endowed with a Riemannian metric of constant curvature 1. The Möbius transformations are the members of $\mathrm{Aut}(\mathbb{P})$, the group of conformal automorphisms of $\mathbb{P}$ under composition. These are intimately connected with both spherical geometry and the geometry of circles. Here are essential facts to note: • An orientation preserving homeomorphism $M$ of the sphere maps circles to circles if and only if $M$ is a Möbius transformation. • In particular, if $P$ is a circle packing in $\mathbb{P}$, then $M(P)$ is a circle packing in $\mathbb{P}$. • If $\{C_1, C_2, C_3\}$ and $\{c_1, c_2, c_3\}$ are any two triples of mutually tangent circles, then there exists a unique Möbius transformation $M$ so that $M(C_j) = c_j$, $j = 1, 2, 3$. • The conformal automorphisms of the unit disc $\mathbb{D}$ and of the complex plane $\mathbb{C}$, $\mathrm{Aut}(\mathbb{D})$ and $\mathrm{Aut}(\mathbb{C})$, are subgroups of $\mathrm{Aut}(\mathbb{P})$. • Aside from the identity $\mathbb{I}$, Möbius transformations all have 1 or 2 fixed points and fall into one of three categories, *parabolic*, *elliptic*, or *hyperbolic*; the parabolic transformations are those with a single fixed point.

It is routine to represent a Möbius transformation $M$ in complex arithmetic as a *linear fractional transformation* $M(z) = (az + b)/(cz + d)$, where $a, b, c, d$ are complex coefficients with $ad - bc \neq 0$. Computationally, we will work with these in the form of $2 \times 2$ complex matrices:

$$M(z) = (az + b)/(cz + d) \text{ is represented by}$$

$$M = \begin{bmatrix} a & b \\ c & d \end{bmatrix}, \text{ with } \det(M) = ad - bc \neq 0.$$

The composition of Möbius transformations is represented by normal matrix multiplication of their matrices, and the inverse of a transformation is represented by the inverse of its matrix. The matrix representation $M$ may be multiplied by any non-zero complex scalar, so we will often normalize to ensure that $ad - bc = 1$. Furthermore, if $M$ is parabolic, we can ensure that $\text{trace}(M) = 2$.

Computations, visualizations, and experiments have been drivers of circle packing since the topic's inception, principally due to the algorithm that Thurston introduced in their 1985 talk. The many refinements of their algorithm now allow the computation of impressively large and complicated complexes, some with millions of circles. These capabilities and connections to conformal geometry have in turn allowed significant applications of circle packing in mathematics [6], in brain imaging [7], physics [8], and engineering [9], not to mention art and architecture.

It is especially important to note the key role that open-ended experiments, visualizations, and serendipity play, even in the purely theoretical aspects of circle packing. The topics in this paper are just the latest examples. Experiments require a laboratory, and for the work here, that laboratory is the open source Java software package `CirclePack`, available on *Github* [10]. All images in this paper and the computations behind them are due to `CirclePack`. Moreover, scripts are available from the author to repeat and extend the experiments.

## 3. Classical, Discrete, and Intrinsic

The Schwarzian derivative was discovered by Lagrange and named after H. Schwarz by Cayley. It is a fundamental Möbius invariant in classical complex analysis, with important applications in topics from function theory, differential equations, and Teichmüler theory, among others. Suppose that $\phi : \Omega \mapsto \Omega'$ is an analytic function between domains $\Omega, \Omega'$ of the complex plane whose derivative $\phi'$ does not vanish. The **Schwarzian derivative** $S_\phi$ is defined by

$$S_\phi(z) = \frac{\phi'''(z)}{\phi'(z)} - \frac{3}{2}\left(\frac{\phi''(z)}{\phi'(z)}\right)^2. \tag{1}$$

There is also a useful **pre-Schwarzian derivative** $s_\phi$:

$$s_\phi(z) = (\ln(\phi'(z)))' = \frac{\phi''(z)}{\phi'(z)} \implies S_\phi(z) = s'_\phi(z) - \frac{1}{2}(s_\phi(z))^2. \tag{2}$$

The Schwarzian derivative is valuable because of its intimate association with Möbius transformations. By direct computation, if $m(z)$ is a Möbius transformation, then $S_m \equiv 0$. The converse also holds: if $S_\phi \equiv 0$ in $\Omega$, then $\phi$ is a Möbius transformation. In general terms; then, *the Schwarzian derivative of a function indicates how far that function differs from being Möbius.* Reinforcing this intuition is the fact that the Schwarzian derivative is invariant under post-composition with Möbius transformations: $S_{m \circ \phi} \equiv S_\phi$. Moreover, for pre-composition, the chain rule gives

$$S_{\phi \circ m}(z) = S_\phi(m(z)) \cdot (m'(z))^2. \tag{3}$$

These features motivate the development of our discrete Schwarzian derivative. This began with the work of Gerald Orick in his Ph.D. thesis [11]. He was searching for a discrete analogue of a classical univalence criterion due to Nehari. Suppose that $\phi$ is an analytic function on the unit disc $\mathbb{D}$. If it were Möbius, then, of course, it would be univalent (i.e., injective). Nehari proved that if $\phi$ is close enough to being Möbius, in the sense $|S_\phi(z)| \le 2/(1 - |z|^2)^2, \forall z \in \mathbb{D}$, then $\phi$ is univalent. Although the search for a discrete version of Nehari's result continues, Orick laid the groundwork for our notion of

Schwarzian derivative, thereby opening a rich vein of questions. (Discretized Schwarzian derivatives have appeared via *cross-ratios* for circle packings with regular square grid or hexagonal combinatorics (see [12] and [13], respectively) and in circle pattern literature (see [14], for example.)

### 3.1. Patches

In concert with the notion of a patch in defining classical conformal structures, a "patch" in a circle packing $P$ will refer to the four circles forming a pair of contiguous faces. Our terminology will be used in both combinatorial and geometric senses. Thus, we will write $\mathfrak{p} = \{v, w \,|\, a, b\}$ for the combinatorial patch formed by faces $f = \{v, w, a\}$ and $g = \{w, v, b\}$ in the complex $K$. We might also use the notation $\mathfrak{p} = \{f \,|\, g\}$. The common edge of the faces is $e = \{v, w\}$, and by convention is positively oriented with respect to the interior of $f$.

The circles of $P$ impose a geometry on $K$, and the corresponding geometric patch in $P$ is $\mathfrak{p} = \{C_v, C_w \,|\, C_a, C_b\}$ forming faces $f$ and $g$ based on the triples $\{C_v, C_w, C_a\}$ and $\{C_w, C_v, C_b\}$, respectively, and with common edge $e = \{C_v, C_w\}$.

Parallel to the classical setting we will also be working with a discrete analytic function $F : P \mapsto P'$ mapping $P$ to a second circle packing $P'$ sharing the complex $K$. For the patch $\mathfrak{p} = \{C_v, C_w \,|\, C_a, C_b\}$ of $P$, we have the corresponding patch $\mathfrak{p}' = F(\mathfrak{p}) = \{C'_v, C'_w \,|\, C'_a, C'_b\}$ of $P'$, and corresponding geodesic triangles $f', g'$ with the shared edge $e'$ of $P'$.

The *discrete Schwarzian derivative* of $F$, denoted $\Sigma_F$, will be a complex function defined on the collection of interior edges $e = \{v, w\}$ of the domain packing $P$. More concretely, the value $\Sigma_F(e)$ will be associated with the tangency point $t_e$ of $C_v$ and $C_w$.

Fix attention on a combinatorial patch $\mathfrak{p} = \{v, w \,|\, a, b\}$ in $K$ with faces $f, g$ in $P$ and $f', g'$ in $P'$, and directed edge $e$. An example is depicted in Figure 3. There exist Möbius transformations $m_f$ and $m_g$ identifying the corresponding faces. We write

$$m_f(f) = f' \quad \text{and} \quad m_g(g) = g'. \tag{4}$$

(A brief note about these equalities: There is a unique Möbius transformation taking the tangent triple $\{C_v, C_w, C_a\}$ to the corresponding triple $\{C'_v, C'_w, C'_a\}$; in practice, it is found by mapping the three tangency points of one to the corresponding tangency points of the other. In hyperbolic and Euclidean settings, these Möbius maps are homeomorphisms of the geodesic triangles formed by the centers of the triples. In spherical geometry, however, Möbius transformations do not necessarily preserve geodesics and circle centers, so the equalities of (4) are symbolic rather that literal.)

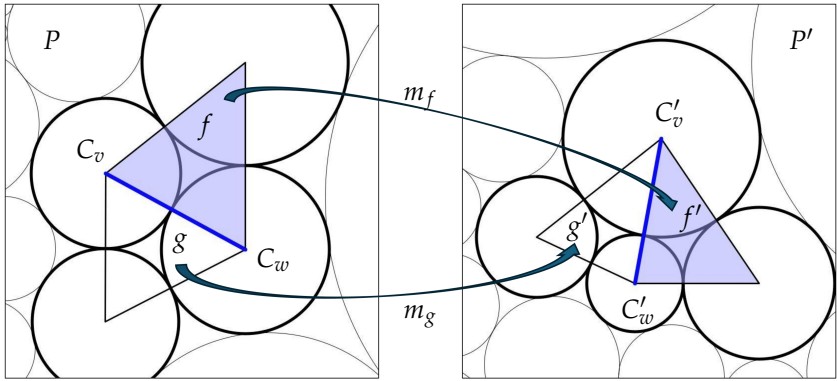

**Figure 3.** The discrete Schwarzian derivative for an edge.

With $m_f$ and $m_g$, we may now define $M_F(e)$ as the Möbius transformation

$$M_F(e) = m_g^{-1} \circ m_f. \tag{5}$$

Though we have adjusted the notation slightly, $M_F(e)$ is the *(directed Möbius) edge derivative* of Orick. The maps $M_F(e)$ have very particular forms observed by Orick. In particular, the definition of $M_F(e)$ shows that it fixes $t_e$, the tangency point of $C_v$ and $C_w$, and that it fixes $C_v$ and $C_w$ themselves as points sets. As a result, if $M_F(e)$ is not the identity, then it is necessarily a parabolic Möbius transformation. We are free to normalize so that $\text{trace}(M_F(e)) = 2$ and $\det(M_F(e)) = 1$. In this case,

$$M_F(e) = \mathbb{I} + \sigma \cdot \begin{bmatrix} t_e & -t_e^2 \\ 1 & -t_e \end{bmatrix}. \tag{6}$$

Moreover, if $\eta = e^{i\theta}$ is the common tangent to $C_v$ and $C_w$ at $t_e$ and is pointing outward from face $f$, then $\sigma$ is a real multiple of its complex conjugate, $\bar{\eta}$.

**Definition 1.** *Let $F : P \longrightarrow P'$ be a discrete analytic function. For each interior edge $e$ of $P$, the value $\sigma$ arising in the computation of $M_F(e)$ as described above is defined as the **(discrete) Schwarzian derivative** of $F$ on the edge $e$ and we write $\sigma = \Sigma_F(e)$.*

There are several properties to observe here:

- Suppose that $-e$ denotes the edge $e$ but with the opposite orientation. Then, $M_F(-e) = (M_F(e))^{-1}$, implying $\Sigma_F(-e) = -\Sigma_F(e)$.
- $\Sigma_F(e) = 0$ if and only if $M_F(e)$ is the identity.
- If $F$ itself were Möbius, then $m_f \equiv m_g \equiv F$ and $M_F(e) = \mathbb{I}$ for every interior edge $e$. Conversely, if $M_F(e) = \mathbb{I}$ for every interior edge $e$, then a simple face-to-face continuation argument would show that $F$ itself is Möbius, with $m_f \equiv F$ for every face $f$.
- Suppose we follow $F$ by a Möbius transformation $m$, say $G \equiv m \circ F : P \to P'' = m(P')$. The new face Möbius maps for $\mathfrak{p}$ are $\widehat{m}_f : f \to m(f') = f''$ and $\widehat{m}_g : g \to m(g') = g''$. Since $\widehat{m}_f = m \circ m_f$ and $\widehat{m}_g = m \circ m_g$, note that

$$M_G(e) = \widehat{m}_g^{-1} \circ \widehat{m}_f = m_g^{-1} \circ m^{-1} \circ m \circ m_f = M_F(e).$$

  The operator $M$ is therefore Möbius invariant. In particular, this implies that the discrete Schwarzian derivative $\Sigma_F$ of a discrete analytic function $F$ displays Möbius invariance like that of the Schwarzian derivative $s_\phi$ of a classical analytic function $\phi$.
- Pre-composition with a Möbius transformation is a different matter. Computations in Appendix A.3 show that the discrete chain rule under pre-composition diverges slightly from the classical rule of (3); see (A10).
- It is very likely that, if a sequence $\{F_n\}$ of discrete analytic functions converges on compacta to a classical analytic function $\phi$, then the sequence $\{\Sigma_{F_n}\}$ also converges on compacta to $S_\phi$. Results of Z–X. He and Oded Schramm in [13] can be used to confirm this for packings with hexagonal combinatorics, but it remains open for more general circle packings.

### 3.2. Intrinsic Schwarzians

The Schwarzian derivative is associated with mappings *between* circle packings. However, we can exploit the same notion in a local sense to provide an "intrinsic schwarzian" for each interior edge of an individual packing. For this we need only consider a target

patch $\mathfrak{p}$ (as occurs within a packing $P$, for instance) and a standard **base patch** $\mathfrak{p}_\Delta$ which we describe next.

The base patch $\mathfrak{p}_\Delta$ consists of two contiguous equilateral triangles, $f_\Delta$ and $g_\Delta$. Here, $f_\Delta$ is formed by the tangent triple of circles of radius $\sqrt{3}$, symmetric about the origin, and having distinguished edge $e_\Delta$ running vertically through $z = 1$. Note that the unit circle is the incircle of $f_\Delta$, that it intersects the edges of $f_\Delta$ at their points of tangency, and that these are the third roots of unity. The tangency point for $e_\Delta$ is $t_e = 1$ and the outward unit vector is $\eta = 1$. The face $g_\Delta$ is an equilateral triangle contiguous along $e_\Delta$, so it shares the two circles of $e_\Delta$ and its third circle is centered at $x = 4$.

A target patch $\mathfrak{p}$ is formed by two triples of circles sharing a pair of circles and has triangular faces we will denote by $f$ and $g$. We do our computation for edge $e_\Delta$ as usual, as though for the mapping $F : \mathfrak{p}_\Delta \to \mathfrak{p}$. That is, we compute the face Möbius transformations $m_f, m_g$, so $m_f(f_\Delta) = f$ and $m_g(g_\Delta) = g$, and then Möbius $M(e_\Delta) = m_g^{-1} \circ m_f$. Referring to (6), note that $\eta = 1$, so $\sigma$ has some real value $s$, and $t_e = 1$, implying

$$M(e_\Delta) = \mathbb{I} + s \cdot \begin{bmatrix} 1 & -1 \\ 1 & -1 \end{bmatrix} = \begin{bmatrix} 1+s & -s \\ s & 1-s \end{bmatrix} = M_s. \tag{7}$$

This matrix is important in the following, so we use the notation $M_s$.

**Definition 2.** *Given a geometric patch $\mathfrak{p}$, the $(2, 1)$-entry (second row, first column) of the Möbius transformation $M_s$ described in (7) is a real value $s$ and is defined as the **(intrinsic) schwarzian** for the shared edge $e$ of $\mathfrak{p} = \{f \mid g\}$.*

The intrinsic schwarzian $s$ completely characterizes the target patch $\mathfrak{p}$ up to Möbius transformations. In particular, it is unchanged if we interchange the labels $f$ and $g$; it is unchanged if $\mathfrak{p}$ is replaced by $m(\mathfrak{p})$ for a Möbius transformation $m$; and if $\mathfrak{p}_1$ and $\mathfrak{p}_2$ are two geometric patches with identical intrinsic schwarzians, then there exists a Möbius transformation $m$ so that $\mathfrak{p}_1 = m(\mathfrak{p}_2)$.

Computations in Appendix A.3 establish the connection between Schwarzian derivatives and intrinsic schwarzians. Given $F : P \longrightarrow P'$, consider a patch $\mathfrak{p} \subset P$, its image patch $\mathfrak{p}' \subset P'$, their edges $e, e'$, respectively, and the Schwarzian derivative $\sigma = \Sigma_F(e)$. Let $s$ and $s'$ denote the intrinsic schwarzians for $e$ and $e'$, respectively. Let $m$ be the Möbius transformation of the base face $f_\Delta$ onto the face $f$ of $\mathfrak{p}$. Computations in the Appendix A.3 show

$$s' = s + \Sigma_F(e) \cdot m'(1) = s + \frac{\Sigma_F(e)}{(c+d)^2}, \text{ where } m(z) = \frac{az+b}{cz+d}, \ ad - bc = 1. \tag{8}$$

As a side note, observe that $\Sigma_F(e) \cdot m'(1)$ is real.

We finish this subsection by illustrating schwarzians in relation to the base patch $\mathfrak{p}_\Delta$. This not only lets the reader gain some intuition, but also leads us to the computational machinery central to the remainder of this paper.

Our base patch $\mathfrak{p}_\Delta$ appears in Figure 4 as the four light blue discs, with $C_b$ being the one on the right. Replacing $C_b$ with some new circle $\widehat{C}$ which is tangent to $C_v$ and $C_w$ forms a new patch. Each new patch leads to some schwarzian $s$ for the edge $e$, so we will denote that new circle by $\widehat{C}_s$. Indeed, by our work above, we have $\widehat{C}_s = M_s^{-1}(C_b)$.

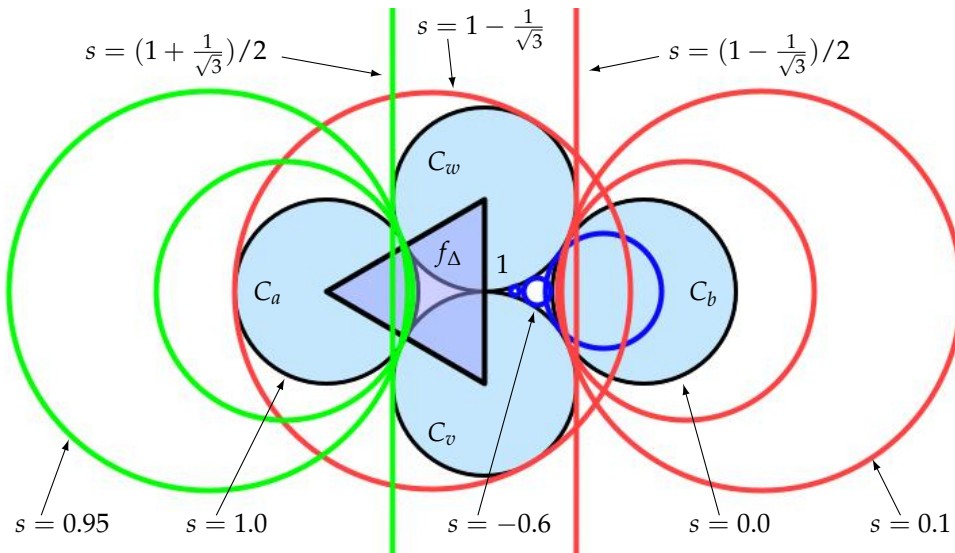

**Figure 4.** Sampling some intrinsic schwarzians.

To explore the various situations, consider these five specific values of $s$:

$$s_0 = 0 \; < \; s_1 = \frac{1 - 1/\sqrt{3}}{2} \; < \; s_2 = 1 - 1/\sqrt{3} \; < \; s_4 = \frac{1 + 1/\sqrt{3}}{2} \; < \; s_5 = 1$$

When $s = s_0 = 0$, $\widehat{C}_s$ is identical to $C_b$. As $s$ decreases through negative values, the corresponding circles $\widehat{C}_s$ become smaller as they drop into the right crevasse between $C_v$ and $C_w$, as shown with some blue examples in Figure 4. As $s$ increases from $s_0$ to $s_2$, the spherical radius of $\widehat{C}_s$ increases, as shown by the red circles. Along the way, when $s = s_1$ then $\widehat{C}_s$ is the line through $\infty$, the vertical line tangent to $C_v$ on $C_w$ on their right. Reaching $s = s_2$, the circle $\widehat{C}_s$ is suddenly tangent to all three of the circles forming $f_\Delta$, but with $\infty$ in its interior. For $s$ larger than $s_2$, the spherical radius of $\widehat{C}_s$ is decreasing, as shown with green examples. The $\widehat{C}_s$ now overlap $C_a$, and on reaching $s_4$, $\widehat{C}_s$ is the vertical line tangent to $C_v$ and $C_w$ on their left. As $s$ grows beyond $s_4$, the circles are moving more deeply into the left crevasse between $C_v$ and $C_w$. Upon reaching $s = s_5$, $\widehat{C}_s$ is identical to $C_a$. This is a critical juncture: the interstices for faces $f$ and $g$ are now reflections of one another through the unit circle. If we let $s$ exceed $s_5 = 1$, then these interstices overlap, a condition we will exclude in later work on branched flowers.

Now, move to the consideration of a generic patch $\mathfrak{p} = \{v, w \,|\, a, b\}$. Suppose the centers and radii for circles $\{c_v, c_w, c_a\}$ forming $f$ and the intrinsic schwarzian for the edge $e = \{c_v, c_w\}$ are known. Then, one can compute the unknown circle $c_b$, and consequently fix the face $g = \{c_w, c_v, c_b\}$. Here are the details:

There exists a Möbius transformation $m_f$ mapping $f_\Delta$ to $f$. Therefore, $m_f^{-1}(f) = f_\Delta$, and so the patch $\widehat{\mathfrak{p}} = m_f^{-1}(\mathfrak{p})$ will be analogous to those depicted in Figure 4. Because schwarzians are invariant under Möbius transformations, the schwarzian for $\widehat{\mathfrak{p}}$ will again be $s$, meaning that its circle $m_f^{-1}(c_b)$ must be $\widehat{C}_s$. Noting that $\widehat{C}_s = M_s(C_b)$, the unknown circle $c_b$ of $\mathfrak{p}$ is given by

$$c_b = (m_f \circ M_s^{-1})(C_b) \quad \text{where} \quad M_s^{-1} = \begin{bmatrix} 1 - s & s \\ -s & 1 + s \end{bmatrix}, \tag{9}$$

a fact we will use extensively in the sequel.

## 4. Packing Layouts

Construction of a circle packing for a given complex *K* typically starts (as Thurston did) with the computation of a *packing label* $R = \{R(v) : v \in K\}$ containing the circle radii. Then comes the *layout* process, the computation of the circle centers. This process utilizes a spanning tree *T* chosen from the dual graph of *K*. Any face $f_0$ of *T* may be designated as the root. Using the radii of its three vertices, one can lay out a tangent triple of circles forming the geometric face $f_0$. For each dual edge $\{f, g\} \in T$, if face *f* is in place, then two of its circles are shared with *g*, and the radius of the remaining circle of *g* is enough to compute its unique position. Thus, starting with the geometric root face $f_0$, one can proceed through *T* to lay out all remaining circles, resulting in the final packing *P*.

This process could instead be carried out using schwarzians, if they were available. Write $S = \{s(e) : e \text{ interior edge}\}$ for the (intrinsic) schwarzians of interior edges for some packing *P*. Starting with *any* (!) tangent triple of circles and identifying it as the base face $f_0$, one can again progress through the edges $\{f, g\}$ of the dual spanning tree *T*. If the face *f* is in place, then using a patch $\mathfrak{p} = \{f \mid g\}$ and the schwarzian $S(e)$ for its shared edge *e*, one can apply (9) to determine the radius and center of the third circle of *g*. Progressing thus through *T* yields a final packing *P* for *K*. Since the whole of *P* is determined by the initial geometric face $f_0$, we can obtain any Möbius image $m(P)$ by starting the layout with the appropriate base face.

There are some issues to address: Using the traditional layout approach *via* radii, the geometry of *P* must be that of the given label *R*. The layout approach *via* intrinsic schwarzians, on the other hand, is by its very nature carried out on the sphere. Indeed, whether the final packing *P* lives in the plane or the hyperbolic plane might well be dictated by the choice of the initial face $f_0$. Perhaps this is the advantage of using schwarzians: one can lay out packings on the sphere or, more generally, on surfaces with projective structures.

I would also point out that when *K* is not simply connected, the layout process, whether with radii or schwarzians, is more subtle; laying out a closed chain of faces which is not null homotopic can lead to non-trivial holonomies, meaning the data is not associated with a circle packing. Let us therefore stick to simply connected complexes *K* for now.

*The Difficulty*

The difficulty in the schwarzian approach lies not with layout, but rather with the computation of the data itself. In introducing circle packing to the world, Thurston also graced us with an iterative algorithm for computing radius data. With radii in hand, one can easily lay out the circles to form *P*. However, their algorithm is restricted to the Euclidean and hyperbolic settings, and despite considerable effort, there is no known algorithm in spherical geometry. There are two key ingredients in Thurston's clever algorithm:

- **Criteria:** Given a label *R* of putative radii, one can directly compute the set $\{\theta_R(v) : v \in K\}$ of associated *angle sums* at the vertices of *K*. *R* is a packing label if and only if $\theta_R(v)$ is an integer multiple of $2\pi$ for every interior *v*.
- **Monotonicities:** There are simple monotonicities in the effects that adjustments in radius labels have on associated angle sums. In particular, a packing label is the zero set of a convex functional, guaranteeing the existence and uniqueness (and computability) of solutions.

It is the monotonicity that fails us in the spherical setting. Building new computational capabilities is the main motivation for looking at schwarzians. That is, we want to replace the data provided by a vertex label $R = \{R(v) : v \in K\}$ with that of an intrinsic schwarzian *edge* label $S = \{S(e) : e \in K\}$.

**Definition 3.** *Let K be a simply connected complex and let S be an edge label, that is, a set of real numbers, one for each interior edge of K. We call S a **packing (edge) label** if there exists a circle packing P on the Riemann sphere whose intrinsic schwarzians are given by S.*

The main question: *What are the packing labels?* Based on experience with radius labels, and in particular, on results in [15], we anticipate that the packing labels will form a $(p-3)$-dimensional differentiable subvariety $\mathfrak{S} \subset \mathbb{R}^k$, where $p$ and $k$ are the numbers of interior vertices and interior edges of $K$, respectively. Describing $\mathfrak{S}$ and more importantly, computing specific packing labels, appears to be very challenging. Our modest approach has been to set up mechanisms for experimentation and discovery. Observations from the experiments in `CirclePack` have led to the clunky but serviceable Theorems 2 and 3 below on packing labels for flowers. As for constructing packing labels for whole complexes, I am less sanguine. Even working with radii data, monotonicity may fail in our spherical setting, and without monotonicity, methods for generating and manipulating packing edge labels will require major new insights and numerical machinery. Edward Crane built an explicit example of a complex $K$ triangulating the sphere with a designated set of its vertices as branch points which has two Möbius inequivalent realizations as circle packings of $\mathbb{P}$. Non-uniqueness is a sobering feature when looking for an algorithm. Nonetheless, let us do what we can and begin by looking at individual flowers.

## 5. Flowers

The search for general packing criteria naturally begins with the study of packing labels for individual flowers. One can easily generate randomized $n$-flowers for any $n$, and thereby obtain a wealth of associated schwarzian labels. Our work, however, lies in the reverse direction: given a label $\{s_0, s_2, \cdots, s_{n-1}\}$, how can one tell if it is a packing label?

Here, we develop and exploit a general process for laying out flowers in a normalized setting. This has been implemented in `CirclePack`, and our investigations have relied on the flexible nature of its computations and visualizations. Our interest lies with closed flowers, and after preliminaries, we work in successive subsections on un-branched flowers, univalent flowers, and finally on branched flowers.

### 5.1. Notation and Preliminaries

In a tangency circle packing, the *flower* of the circle $C = C_v$ for an interior vertex is denoted $\{C; c_0, c_1, \cdots, c_{n-1}\}$, where $c_0, \cdots, c_{n-1}$ is the chain of *petals* which wrap around $C$ with the last tangent to the first. The ordered chain of interior edges emanating from $v$ may be written as $\{e_0, e_1, \cdots, e_{n-1}\}$, where $e_j$ is the edge $\{C, c_j\}$ and hence is the shared edge in the patch $\mathfrak{p} = \{c_j, C \,|\, c_{j-1}, c_{j+1}\}$. We write $\{s_0, s_1, \cdots, s_{n-1}\}$ for the corresponding intrinsic schwarzians, a packing label for this flower. (Note that the indexing for $n$-flowers is always modulo $n$.)

Putting the first triple of circles, $\{C, c_{n-1}, c_0\}$, in place, one can then use schwarzians $s_0, s_1, \cdots, s_{n-3}$ in succession to place $c_1, c_2, \cdots, c_{n-2}$ and possibly complete the geometric flower. But is this a flower? Some conditions must be needed since this procedure did not even utilize the given schwarzians $s_{n-2}$ or $s_{n-1}$. To be a packing label, the layout must avoid three potential incompatibilities:

(a) The flower may fail to close; that is, $c_{n-1}$ may fail to be tangent to $c_0$.
(b) The patch $\mathfrak{p}_{n-1} = \{c_{n-1}, C \,|\, c_{n-2}, c_0\}$ may fail to have schwarzian $s_{n-1}$.
(c) The patch $\mathfrak{p}_0 = \{c_0, C \,|\, c_{n-1}, c_1\}$ may fail to have schwarzian $s_0$.

Since flowers and their schwarzians are unchanged under Möbius transformations, one can choose any convenient normalization. We have chosen that illustrated in Figure 5 (in this instance, a 7-flower). This approach sidesteps the numerical difficulties working with

infinity. The notations of the figure are those we will use throughout the paper: the upper half plane represents the central circle $C$, the half plane $\{z = x + iy : y \leq -2\}$ represents $c_0$ (so the tangency between $C$ and $c_0$ lies at infinity), and the petal $c_1$ of radius 1 is tangent to $C$ at $t_1 = 0$. The successive petals $\{c_2, \cdots, c_{n-1}\}$ have tangency points $\{t_2, \cdots, t_{n-1}\}$. The successive *Euclidean radii* will be denoted by $\{r_2 \cdots, r_{n-1}\}$, and successive *displacements* by $\delta_j = t_{j+1} - t_j$.

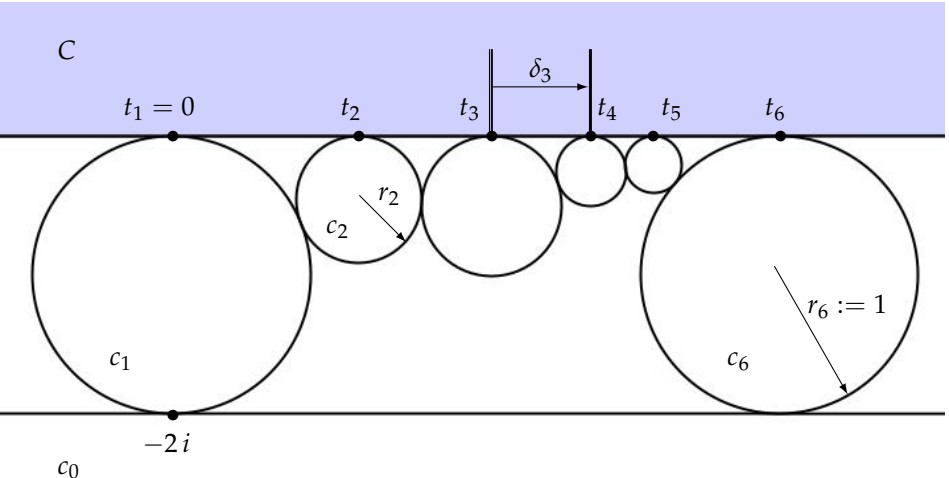

**Figure 5.** Notations for normalized flower layouts.

Figure 6 provides a sampler of normalized flowers. In Figure 6a, the non-contiguous petals $c_3$ and $c_0$ overlap. Figure 6b illustrates an "extraneous tangency", as petals $c_{j-1}$ and $c_{j+1}$ are tangent, even though they are not neighbors in the flower structure. Figure 6c illustrates a flower whose seven petals reach twice about $C$; necessarily, some of them overlapping one another. Note that Figures 5 and 6d are univalent flowers, Figure 6a is non-univalent, but is un-branched, while Figure 6c is branched. Figure 6d illustrates an extremal situation among univalent flowers: petals $c_2, \cdots, c_5$ all have extraneous tangencies with $c_0$, yet the petals' interiors are mutually disjoint. This illustrates the configuration among normalized univalent $n$-flowers with the greatest distance between the end petals $c_1$ and $c_{n-1}$.

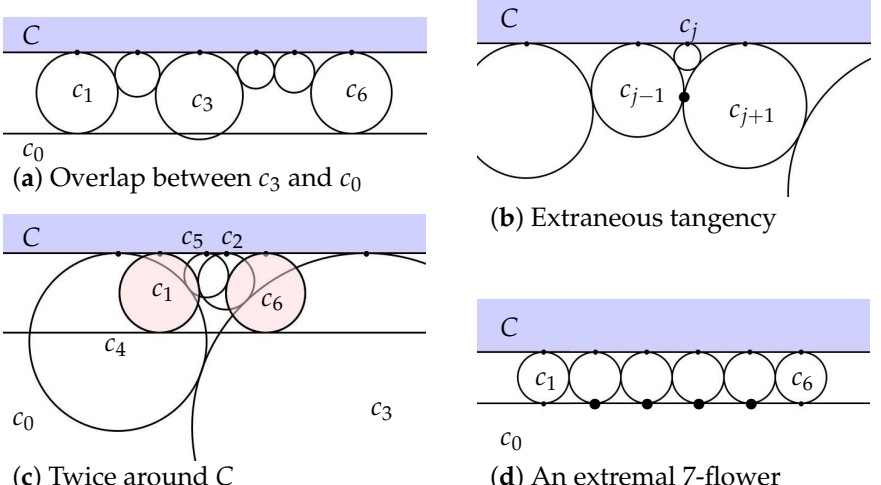

**(a)** Overlap between $c_3$ and $c_0$

**(b)** Extraneous tangency

**(c)** Twice around $C$

**(d)** An extremal 7-flower

**Figure 6.** Examples of normalized flower variety.

*5.2. Flower Layouts*

Given putative schwarzians $\{s_0, s_1, \cdots, s_{n-1}\}$ for an $n$-flower, the associated flower can be laid out in normalized form using the following mechanical process, which relies on computations carried out in Appendix A.1.

(1)  We start with the half planes $C$ and $c_0$ and the circle $c_1$ in their normalized positions.
(2)  With $c_1$ in place and given the schwarzian $s_1$ for its edge, the formulas of (A2) in Appendix A.1 yield the tangency point $t_2$ and radius $r_2$ of $c_2$.
(3)  With $c_1$ and $c_2$ in place and given $s_2$, we are in the "generic" Situation 3 of Appendix A.1, but in the special case where $r = 1$. In particular, we can place $c_3$ by using (A3) to compute radius $r_3$ and displacement $\delta_2$, leading to tangency point $t_3$.
(4)  Hereafter, we remain in the generic Situation 3, so we can place the remaining petals by inductively applying (A3) to compute radii $r_j$ and the displacements to determine the tangency points $t_j, j = 4, \cdots, n-1$.

At this point, we would have the petals of the presumptive flower all in place. However, we can see concretely how the compatibility conditions mentioned earlier might fail:

(a)  If $r_{n-1} \neq 1$, $c_{n-1}$ fails to be tangent to $c_0$—and the flower does not close up.
(b)  If $t_{n-1} - t_{n-2} \neq 2/(\sqrt{3}(1 - s_{n-1}))$ after the final application of (A3) would mean that $s_{n-1}$ is not the schwarzian for patch $\{c_{n-1}, C \mid c_{n-2}, c_0\}$.
(c)  If $t_{n-1} \neq 2\sqrt{3}(1 - s_0)$ then (A1) tells us that $s_0$ is not the schwarzian for patch $\{c_0, C \mid c_{n-1}, c_1\}$.

Therefore, our work, both theoretical and numerical, depends on a modification of this process:

**Example 1.** *Treating the $n - 3$ schwarzians $s_1, \cdots, s_{n-3}$ as parameters, we build the normalized flower as described above up to and including the layout of $c_{n-2}$. We then **force** closure by setting $r_{n-1} = 1$ and placing the last petal $c_{n-1}$ tangent to $c_{n-2}$.*

This Layout Process underlies all the work in this section. Once the construction has put all petals in place, one can **directly compute** the three remaining schwarzians $s_{n-2}, s_{n-1}$, and $s_0$ to fill out the full packing label $\{s_0, \cdots, s_{n-1}\}$.

**Theorem 1.** *Given schwarzians $\{s_1, \cdots, s_{n-3}\}$, the **Layout Process** results in a legitimate $n$-flower except in the two following situations: (a) when $c_{n-2}$ is tangent to $C$ at infinity or (b) when the computed $s_0$ exceeds 1.*

**Proof.** The first statement requires no proof, as the mechanics are straightforward. As for situation (a), in this case $c_{n-2}$ is a half plane, meaning that placement of $c_{n-1}$ simultaneously tangent to $C, c_{n-2}$, and $c_0$ is either impossible or ambiguous. Situation (b) violates a condition we placed on schwarzians; in this case, petal $c_{n-1}$ ends up to the left of $c_1$. For details on the exceptions, go to the closing paragraph of Appendix A.1.  □

*5.3. Important Observations*

In the pencil-and-paper computations leading to the formulas of the Appendix A (and the associated code in `CirclePack`), it became clear that a new parameter $u = 1 - s$ is preferrable to the schwarzian $s$ itself. Instead of label $\{s_0, \cdots, s_{n-1}\}$, we will interchangeably use $\{u_0, \cdots, u_{n-1}\}$, though we continue to treat the $s$-variables as the proper labels. The author can offer no geometric significance for this new $u$-variable, but converting the $s$'s in our discussion of Figure 4 to $u$'s may help the reader develop some intuition. For the reasons discussed there, we limit our work to $s \in (-\infty, 1)$, and thus $u \in (0, \infty)$.

Also note that, although our normalization picks $c_0$ to be a half plane, any petal of a flower may be designated as $c_0$ as a simple matter of indexing. Furthermore, if the order of the petals in a flower is reversed, the result is still a flower and the order of schwarzians will have been reversed. These observations explain, respectively, the cyclic and the symmetric features in this lemma.

**Lemma 1.** *Suppose that $\{s_0, \cdots, s_{n-1}\}$ is a packing (edge) label for an n-flower. If one shifts the order of the schwarzians cyclically or reverses the order, the result is again a packing label. This holds equally for the u-variables $\{u_0, \cdots, u_{n-1}\}$.*

*5.4. Un-Branched Flowers*

Our results are most complete in the case of un-branched $n$-flowers, where we work step by step starting with $n = 3$. Examples for degrees $n = 3$, 4, 5, and 6 are shown in Figure 7. These provide some visual clues to the patterns we will discuss below. Figure 7d is also cautionary, as it illustrates three Möbius equivalent normalized representations of the same 6-flower, differing only by which petal had been designated as $c_0$.

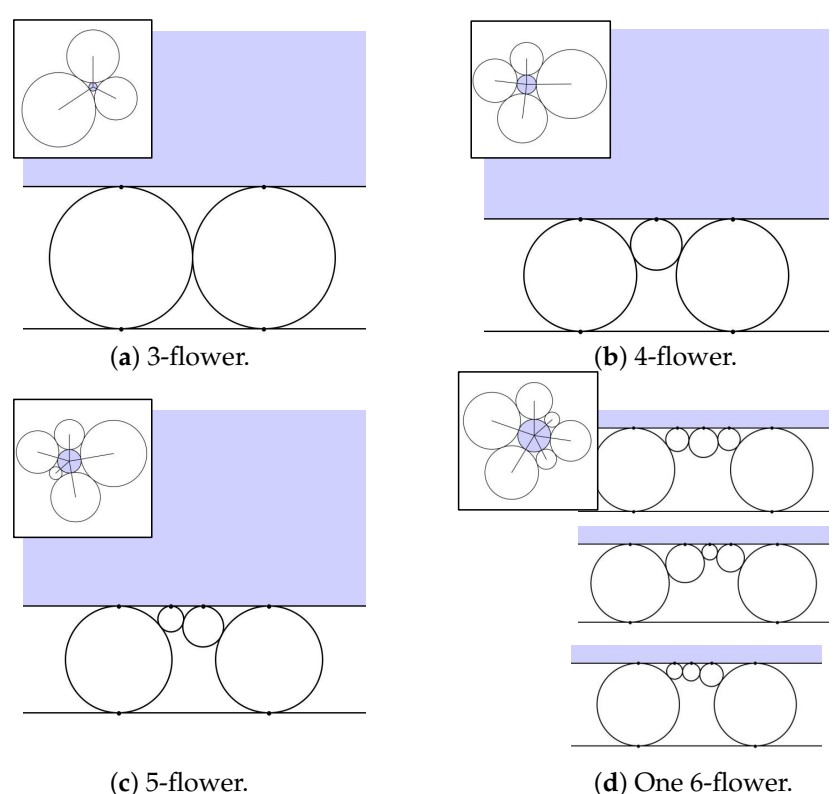

(**a**) 3-flower.　　　　　　　　(**b**) 4-flower.

(**c**) 5-flower.　　　　　　　　(**d**) One 6-flower.

**Figure 7.** Normalized flowers.

5.4.1. 3-Flowers

There is only one 3-flower up to Möbius transformations. In particular, the three edge schwarzians share identical values. Note in Figure 4 that the value $s = 1 - 1/\sqrt{3}$ leads to a 3-flower, namely that formed by $C_v$, $C_a$, and the associated $\widehat{C}_s$ (enclosing $\infty$).

**Lemma 2.** *The intrinsic schwarzian for any edge of a 3-flower is $s = 1 - 1/\sqrt{3}$, and hence, $u = 1/\sqrt{3}$.*

This value of $s$ may also occur in higher degree flowers in the case of extraneous tangency, as seen, for example, in Figure 6b, where the schwarzian $s_j$ takes this value. (It is

worth noting that in circle packings, interior vertices of degree 3 are somewhat extraneous themselves: the associated circle is determined uniquely by its three neighboring circles and could be omitted without affecting the packing's overall geometric structure. On the other hand, omitting such a circle does affect schwarzians, namely those of the outer three edges of its flower.)

5.4.2. 4-Flowers

Figure 8 shows a sequence of un-branched 4-flowers. Petals $c_1$ and $c_3$ have a radius of 1, so it is clear that the size of the shaded petal, $c_2$, determines the entire normalized 4-flower. There is thus one degree of freedom. In (A2) of Appendix A.1, we see the radius of $c_2$ as a monotone function of $s_1$, so we can use $s_1$ to parameterize all 4-flowers. (The curious case of a branched 4-flower will be displayed in Section 5.6.)

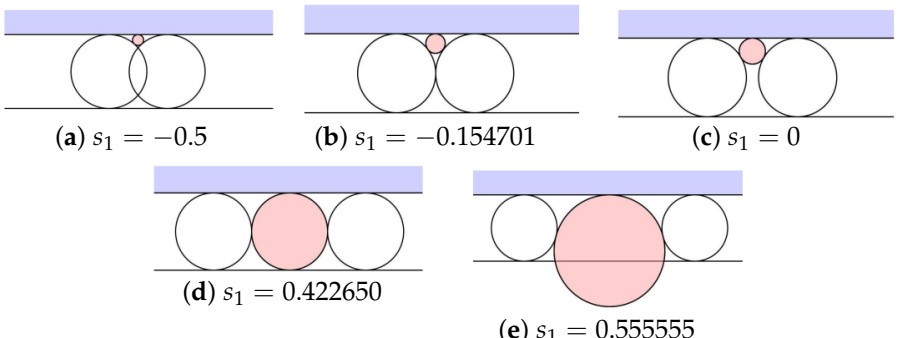

**(a)** $s_1 = -0.5$      **(b)** $s_1 = -0.154701$      **(c)** $s_1 = 0$

**(d)** $s_1 = 0.422650$

**(e)** $s_1 = 0.555555$

**Figure 8.** Variety in 4-flowers.

- Apply (A2) to compute the tangency point and radius of $c_2$:

$$\delta_1 = t_2 = 2/(\sqrt{3}\, u_1) \quad \text{and} \quad r_2 = 1/(\sqrt{3}\, u_1)^2.$$

- Apply (A4) with $R = r_2$ and $r = 1$ to compute $u_2 = 2/(3u_1)$.

Here, we initiate a pattern that we will carry forward for larger $n$: namely we define functions

$$\mathfrak{U}_4(x) = 2/(3x), \qquad \mathfrak{C}_2(x) = x,$$

and note that $u_2 = \mathfrak{U}_4(u_1)$ under the constraint that $\mathfrak{C}_2(u_1) > 0$. Furthermore, as we will do for larger degrees, we can engage Lemma 1. With successive left shifts of the parameters we conclude that $u_3 = \mathfrak{U}_4(u_2)$ and $u_0 = \mathfrak{U}_4(u_3)$, thereby completing the full packing label. In particular, note that $u_1 = u_3$, $u_0 = u_2$, and $u_1 u_2 = 2/3$. We arrive at a very clean characterization of packing labels for 4-flowers:

**Lemma 3.** *Every un-branched 4-flower has edge schwarzians of the form $\{s, s', s, s'\}$ where $s$ and $s'$ satisfy $(1-s)(1-s') = 2/3$ (i.e., $uu' = 2/3$). Moreover, the 4-flower is univalent if and only if $s$ and $s'$ lie in the closed interval $I = [1 - \frac{2}{\sqrt{3}}, 1 - \frac{1}{\sqrt{3}}]$ (i.e., $u, u' \in [\frac{1}{\sqrt{3}}, \frac{2}{\sqrt{3}}]$).*

Figure 8b–d are univalent 4-flowers, with (b) and (d) representing the extremes of the parameter values allowed for univalence. One should not be misled by this complete understanding of 4-flowers—things become increasingly more complicated as the degree goes up.

### 5.4.3. Five-Flowers

Our Layout Process tells us that we have 2 degrees of freedom, namely $s_1$ and $s_2$. Figure 9 indicates the quantities we can compute as functions of these for a generic 5-flower.

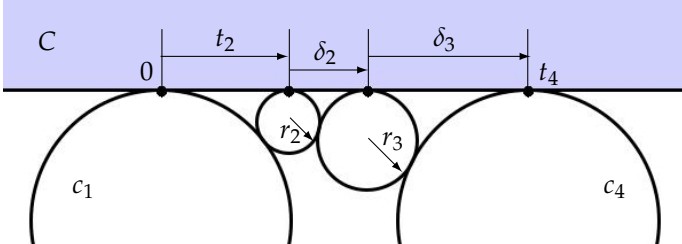

**Figure 9.** Quantities to compute in a normalized 5-flower.

Quantities resolve cumulatively as we add petals: with $C, c_0$, and $c_1$ in place, we can apply the computation in the earlier 4-degree case to compute $r_2 = 1/(\sqrt{3}\,u_1)^2$. From there, successive computations from Appendix A.1 yield various radii and displacements:

- Applying (A3) with $u = u_2$, $r = 1$, and $R = r_2$:

$$\delta_2 = \frac{2}{\sqrt{3}\,u_1(3u_1u_2 - 1)} \quad \text{and} \quad r_3 = \frac{1}{(3u_1u_2 - 1)^2}.$$

Here, we encounter a constraint: if $(3u_1u_2 - 1) <= 0$, then $\delta_2$ will be negative or undefined. As we will see later, this can happen for branched flowers. For the un-branched case, we must impose the condition $(3u_1u_2 - 1) > 0$.

- Using radii $r_3$ from above and the mandated $r_4 = 1$:

$$\delta_3 = 2\sqrt{r_3} = \frac{2}{3u_1u_2 - 1},$$

- Setting $r = r_2$ and $R = r_3$ in (A4) implies

$$u_3 = \frac{u_1 + 1/\sqrt{3}}{3u_1u_2 - 1}.$$

Define the rational function $\mathfrak{U}_5$ and the polynomial $\mathfrak{C}_3$ as follows:

$$\mathfrak{U}_5(x_1, x_2) = \frac{x_1 + 1/\sqrt{3}}{3x_1x_2 - 1}, \qquad \mathfrak{C}_3(x_1, x_2) = 3x_1x_2 - 1.$$

The computation of $u_3$ becomes simply $u_3 = \mathfrak{U}_5(u_1, u_2)$ under the constraint $\mathfrak{C}_3(u_1, u_2) > 0$. Applying the cyclic property of Lemma 1, we get in succession $u_4 = \mathfrak{U}_5(u_2, u_3)$ and $u_0 = \mathfrak{U}_5(u_3, u_4)$.

### 5.4.4. 6-Flowers

We work through this one additional case because the full strength of Situation 3 and (A3) is first felt with the addition of the sixth petal. (Also because 6-flowers have always occupied a prominent place in circle packing: in the "curvature" language common in this topic, 6-flowers are "flat".)

Six-flowers involve three degrees of freedom with parameters $\{s_1, s_2, s_3\}$. We may extend the notations in the previous section and Figure 9 by adding one additional petal. The computations of $t_2$ and $\delta_2$ and the constraint $(3u_1u_2 - 1) > 0$ are exactly as earlier. The computation for $\delta_3$, however, needs to be revisited.

- Applying (A3) with $r_2$ and $r_3$ as computed earlier gives

$$\delta_3 = \delta(u_3, r_2, r_3) = \frac{2}{\sqrt{3}\,(3u_1 u_2 - 1)(3u_1 u_2 u_3 - u_1 - u_3)},$$

$$r_4 = \frac{\delta_3^2}{4r_3} = \frac{1}{3(3u_1 u_2 u_3 - u_1 - u_3)^2}.$$

Note that $1/\sqrt{r_4} = \sqrt{3}(3u_1 u_2 u_3 - u_1 - u_3)$ if this is positive. Then, by completing the flower with petal $c_5$ of mandated radius 1, we can compute the next label:

- Applying (A4) with $r = r_3$ and $R = r_4$ gives

$$u_4 = \frac{u_1 u_2}{(3u_1 u_2 u_3 - u_1 - u_3)}.$$

Define the rational function $\mathfrak{U}_6$ and polynomial $\mathfrak{C}_4$:

$$\mathfrak{U}_6(x_1, x_2, x_3) = x_1 x_2 / (3x_1 x_2 x_3 - x_1 - x_3),$$
$$\mathfrak{C}_4(x_1, x_2, x_3) = \sqrt{3}(3x_1 x_2 x_3 - x_1 - x_3).$$

Thus, $u_4 = \mathfrak{U}_6(u_1, u_2, u_3)$ under the assumptions $\mathfrak{C}_3(u_1, u_2) > 0$ and $\mathfrak{C}_4(u_1, u_2, u_3) > 0$. Cyclic shifts provide the remaining two labels:

$$u_5 = \mathfrak{U}_6(u_2, u_3, u_4) \quad \text{and} \quad u_0 = \mathfrak{U}_6(u_3, u_4, u_5).$$

5.4.5. The General Case

In the $n$-flower case, we are starting with $n - 3$ parameters $\{u_1, \cdots, u_{n-3}\}$. A look at the various formulas from Appendix A.1 suggests focusing the on reciprocal roots of the radii. Here are the first few expressions. (We introduce a convenient notational devise that abbreviates a product of $u$'s via multiple subscripts, allowing us, for example, to write $u_{1,4,5}$ in place of the product $u_1 u_4 u_5$.)

$$\begin{aligned}
1/\sqrt{r_2} &= \mathfrak{C}_2(u_1) = \sqrt{3}u_1, \\
1/\sqrt{r_3} &= \mathfrak{C}_3(u_1, u_2) = 3u_{1,2} - 1, \\
1/\sqrt{r_4} &= \mathfrak{C}_4(u_1, u_2, u_3) = \sqrt{3}(3u_{1,2,3} - u_1 - u_3), \\
1/\sqrt{r_5} &= \mathfrak{C}_5(u_1, \cdots, u_4) = 9u_{1,2,3,4} - 3u_{1,4} - 3u_{3,4} - 3u_{1,2} + 1, \\
1/\sqrt{r_6} &= \mathfrak{C}_6(u_1, \cdots, u_5) = \\
&\quad \sqrt{3}(9u_{1,2,3,4,5} - 3u_{1,4,5} - 3u_{3,4,5} - 3u_{1,2,5} - 3u_{1,2,3} + u_1 + u_3 + u_5).
\end{aligned} \tag{10}$$

$\cdots\cdots$

For a given $j$, the expression for $1/\sqrt{r_j}$ is ensured only if $\mathfrak{C}_k(u_1, \cdots, u_{k-1}) > 0$ for $k = 2, \cdots, j$, and only within $n$-flowers for which $n \geq (j + 2)$. Should $\mathfrak{C}_j$ be negative for some $j$, then the flower will be branched.

With these cautions in mind, the functional notations can become quite convenient. A label for an $n$-flower may be expressed as an $n$-vector $p = (u_0, \cdots, u_{n-1})$. We may now write $\mathfrak{C}_j(u_1, \cdots, u_{j-1})$ as $\mathfrak{C}(p)$, noting that $\mathfrak{C}_j$ uses only the $j - 1$ coordinates $1, 2, \cdots, (j-1)$ of $p$. Rewriting (A3) in functional notation, we have

$$\mathfrak{C}_{j+1}(p) = \sqrt{3}u_j \mathfrak{C}_j(p) - \mathfrak{C}_{j-1}(p), \quad 3 \leq j \leq n - 3. \tag{11}$$

When our construction places the last petal, $c_{n-1}$, we compute $u_{n-2}$ by applying (A4). In functional notation, this becomes

$$u_{n-2} = \mathfrak{U}_n(p) = \frac{1 + \mathfrak{C}_{n-3}(p)}{\sqrt{3}\,\mathfrak{C}_{n-2}(p)}, n \geq 5,$$

where $\mathfrak{U}_n$ depends only on coordinates $1, \cdots, (n-3)$ of $p$. Here are several of these functions in explicit form:

$$
\begin{aligned}
u_2 &= \mathfrak{U}_4(u_1) = \frac{2}{3u_1}, \\
u_3 &= \mathfrak{U}_5(u_1, u_2) = \frac{u_1 + 1/\sqrt{3}}{3u_{1,2} - 1}, \\
u_4 &= \mathfrak{U}_6(u_1, u_2, u_3) = \frac{u_{1,2}}{3u_{1,2,3} - u_1 - u_3}, \\
u_5 &= \mathfrak{U}_7(u_1, u_2, u_3, u_4) = \frac{3(3u_{1,2,3} - u_1 - u_3) + 1/\sqrt{3}}{3(3u_{1,2,3,4} - u_{1,2} - u_{1,4} - u_{3,4}) + 1}, \\
u_6 &= \mathfrak{U}_8(u_1, u_2, u_3, u_4, u_5) \\
&= \frac{3(3u_{1,2,3,4} - u_{1,2} - u_{1,4} - u_{3,4}) + 2}{3(9u_{1,2,3,4,5} - 3u_{1,2,3} - 3u_{1,2,5} - 3u_{1,4,5} - 3u_{3,4,5} + u_1 + u_3 + u_5)}.
\end{aligned}
\tag{12}
$$

$\cdots \cdots$

We gather the results for un-branched flowers in this theorem. Also see the comments that follow.

**Theorem 2.** *Given $n > 3$, the parameters $\{u_1, \cdots, u_{n-3}\}$ are part of a packing label for an un-branched n-flower if and only if*

$$\mathfrak{C}_j(u_1, \cdots, u_{j-1}) > 0, \; j = 2, \cdots, (n-2). \tag{13}$$

*In this case, these expressions*

$$
\begin{aligned}
u_{n-2} &= \mathfrak{U}_n(u_1, \cdots, u_{n-3}), \\
u_{n-1} &= \mathfrak{U}_n(u_2, \cdots, u_{n-2}), \\
u_0 &= \mathfrak{U}_n(u_3, \cdots, u_{n-1}),
\end{aligned}
\tag{14}
$$

*allow the computation of the three remaining labels.*

This Theorem simultaneously provides a characterization, a parameterization, and a computational tool for un-branched flowers. Here are some observations:

(i)   The functions $\mathfrak{U}_n$ and $\mathfrak{C}_j$ are particularly valuable in light of Lemma 1. (Remember, these labels are cyclic mod($n$).) In a packing label $\{u_0, \cdots, u_{n-1}\}$, any one of its entries $u_j$ may be written as $u_j = \mathfrak{U}_n(\sigma)$, where $\sigma$ is the sequence of $n-3$ entries preceding $u_j$ (or the reverse of the $n-3$ entries following $u_j$).

(ii)  Additional relationships pertaining to the normalized flower may be extracted from the formulas of Appendix A.1. Here are some examples:

$$\frac{1}{\sqrt{r_j}} = \frac{\sqrt{3}\,u_{j-1}}{\sqrt{r_{j-1}}} - \frac{1}{\sqrt{r_{j-2}}}, \; j = 3, 4, \cdots, n-1. \tag{15}$$

$$u_j = \frac{\sqrt{\frac{r_j}{r_{j-1}}} + \sqrt{\frac{r_j}{r_{j+1}}}}{\sqrt{3}}, \; j = 2, \cdots, n-3.$$

$$\delta_j = 2\sqrt{r_j r_{j+1}}, \; j = 1, \cdots, n-2.$$

(iii) A careful look at the formulas for a normalized flower will show that the radii $r_j$, reciprocal roots $1/\sqrt{r_j}$, and tangency points $t_j$ of the petals are all rational functions of the $u$-parameters (likewise for the $s$-parameters). Moreover, these rational functions have their coefficients in the number field $\mathbb{Q}[\sqrt{3}]$.

(iv) The rational functions $\mathfrak{U}_n$ and polynomials $\mathfrak{C}_j$ also have coefficients in $\mathbb{Q}[\sqrt{3}]$. Note that, for each $n$, $\mathfrak{C}_{n-2}(u_1, \cdots, u_{j-3})$ is a pole of $\mathfrak{U}_n(u_1, \cdots, u_{n-3})$. On a practical note, unlike expressions such as (10), the functions $\mathfrak{U}.$ and $\mathfrak{C}.$ are independent of the flower normalization.

(v) The functions $\mathfrak{U}.$ have intriguing self-referential behavior under cyclic shifts and reversals, and this would seem to make them quite special. For example, these expressions show how the $\mathfrak{U}_n$ can be nested; here, $\vec{u}_{j,k}$ denotes the sequence $\{u_j, \cdots, u_k\}$.

$$\begin{aligned}
u_{n-2} &= \mathfrak{U}_n(\vec{u}_{1,n-3}) \\
u_{n-1} &= \mathfrak{U}_n(\vec{u}_{2,n-3}, \mathfrak{U}_n(\vec{u}_{1,n-3})) \\
u_0 &= \mathfrak{U}_n(\vec{u}_{3,n-3}, \mathfrak{U}_n(\vec{u}_{1,n-3}), \mathfrak{U}_n(\vec{u}_{2,n-3}, \mathfrak{U}_n(\vec{u}_{1,n-3}))).
\end{aligned} \tag{16}$$

The explicit expressions would be quite messy, but would express the labels $u_{n-2}, u_{n-1}, u_0$ of (14) directly as functions of the parameters $u_1, \cdots, u_{n-3}$.

We are now in a position to describe the parameter space for un-branched $n$-flowers using vectors $p = (u_0, u_2, \cdots, u_{n-1})$ in $\mathbb{R}^n_+$. Define $\mathcal{V}_n$ as the common solutions of the three rational expressions of (16). In particular, $\mathcal{V}_n$ is an algebraic variety of the dimension $n-3$ over the number field $\mathbb{Q}[\sqrt{3}]$. There are restrictions, however, as $p$ must reside in the set $\mathcal{C}$ where components $u_j$ are positive and the $\mathcal{C}_j$ satisfy (13). The parameter space for un-branched $n$-flowers is thus $\mathcal{F}_n = \mathcal{V}_n \cap \mathcal{C} \subset \mathbb{R}^n$.

The parameter space $\mathcal{F}_n$ has some rather unique features. Each point $p \in \mathcal{F}_n$ determines a unique flower (that is, unique up to Möbius transformations). On the other hand, each un-branched $n$-flower is associated with up to $n$ distinct points $p$, since by Lemma 1, one can cyclically permute (the coordinates of) $p$. We have treated $\{u_1, \cdots, u_{n-3}\}$ as the independent variables, but in fact, any $n-3$ cyclically successive coordinates can take on this role. One might wonder about the description of $\mathcal{C}$, defined in terms of inequalities depending on $u_1, \cdots, u_{n-3}$: most of the individual inequalities in (13) would fail under the cyclic permutation of their arguments. However, if **all** of the inequalities hold, then $p \in \mathcal{F}_n$, and as a result, each of them individually holds under cyclic permutation. Likewise, reversing the coordinates of $p \in \mathcal{F}_n$ gives a (generically distinct) point of $\mathcal{F}_n$.

*5.5. Univalent Flowers*

Among the *un-branched* flowers are the *univalent* flowers, those whose petals have mutually disjoint interiors. In the study of discrete analytic functions, univalent flowers are (along with branched flowers) the most important. Define the subset $\mathcal{U}_n \subset \mathcal{F}_n$ to consist of parameters associated with univalent $n$-flowers.

We develop two collections of inequalities which together characterize the points of $\mathcal{U}_n$. The inequalities of this first collection are very easy to check.

$$\frac{1}{\sqrt{3}} \le u_j \le \frac{(n-2)}{\sqrt{3}}, \; j = 0, \cdots, n-1. \tag{17}$$

The second collection (18) of inequalities depends on putting the flowers in their normalized setting, and checking these takes more work because a point $p \in \mathcal{F}_n$ has $n$ normalized layouts based (as always) on which petal is designated as "$c_0$". As a result, the indexing used in laying out a flower—the indexing occurring in various formulas—will generally disagree with the official indexing of the entries in the given $p$. We introduce a notational device to more efficiently state the inequalities. Notation: Use $\vec{p}^{\,k}$ to indicate a cyclic permutation of $p$ which shifts the original coordinate $u_k$ to become $u_0$. Note that $p \in \mathcal{U}_n$ if and only if $\vec{p}^{\,k} \in \mathcal{U}_n$ for all $k = 0, \cdots, n-1$. In these inequalities, $r_j(p)$ denotes the radius of the $j$th petal circle in the normalized layout for $p$.

$$r_j(\vec{p}^{\,k}) \leq 1, \ \forall j = 2, \cdots, n-2 \text{ and } \forall k = 0, \cdots, n-1. \tag{18}$$

**Theorem 3.** *Given $n > 3$, suppose that $p$ represents an un-branched $n$-flower, so $p \in \mathcal{F}_n$. Then, $p$ represents a univalent $n$-flower, $p \in \mathcal{U}_n$, if and only if $p$ satisfies the inequalities of (17) and (18).*

**Proof.** *Necessity:* Suppose that $p \in \mathcal{U}_n$. We observed in discussing Figure 4 that, if $s > (1 - \frac{1}{\sqrt{3}})$, then $\widehat{C}_s$ would overlap $C_a$, contradicting univalence. This gives the lower bound of (17). The upper bound depends on $n$. Consider the tangency point $t_{n-1}$ in the normalized flower. It is clear that the largest $t_{n-1}$ could be for a univalent flower that occurs for "extremal" flowers like that illustrated (for $n = 7$) in Figure 6d. In this case, $t_{n-1} = 2(n-2)$. Since our flower is univalent, $t_{n-1} \leq 2(n-2)$. On the other hand, by (A1), $2\sqrt{3}u_0 = t_{n-1}$. We conclude that $u_0 \leq \frac{(n-2)}{\sqrt{3}}$. Lemma 1 tells us that any $u_j$ can be treated as $u_0$, so $u_j \leq \frac{n-2}{\sqrt{3}}$.

We prove (18) by contradiction: suppose (18) fails for some $k$ and $3 \leq j \leq n-1$. Then in the shifted indexing of $\vec{p}^{\,k}$, the petal circle $c_j$, having radius greater than 1 would necessarily overlap the half plane $c_0$, contradicting univalence. We have established the necessity of (17) and (18).

*Sufficiency:* Suppose the point $p \in \mathfrak{F}_n$ satisfies (17) and (18) but its flower is not univalent. That is, suppose there exists some pair $c_j, c_k$ of petal circles that overlap, $0 \leq j < k \leq n-1$. Suppose first that there is a single petal between $c_j$ and $c_k$, so $k = j+2$. In the normalized layout for the shifted point $\vec{p}^{\,j}$, the two end circles of the layout would overlap, implying by (A1) that $2\sqrt{3}\,u_j < 2$ and hence $u_j < 1/\sqrt{3}$, violating (17). On the other hand, if $c_j$ and $c_k$ are separated by at least two petals, then with the new indexing of $\vec{p}^{\,j}$, the petal $c_j$ becomes $c_0$, and $c_k$ becomes a petal strictly between $c_1$ and $c_{n-1}$. Since that petal overlaps $c_0$, its radius must exceed 1, contradicting (18). This completes the proof of sufficiency. $\square$

In practice, (17) is trivial to check, while (18) takes most of the work. Although the normalized flowers for shifts $\vec{p}^{\,k}$ are all Möbius images of one another, one must still check (18) for each of the normalizations in turn. In a given normalization, one can use the quasi-recursive expression (15) along with the fact that $r_0 = \infty$ and $r_1 = 1$ to quickly see whether an instance of (18) is violated.

*5.6. Branched Flowers*

Now we address the more difficult setting of branched flowers, which are essential in the study of discrete analytic functions. Figures 10 and 11 provide examples.

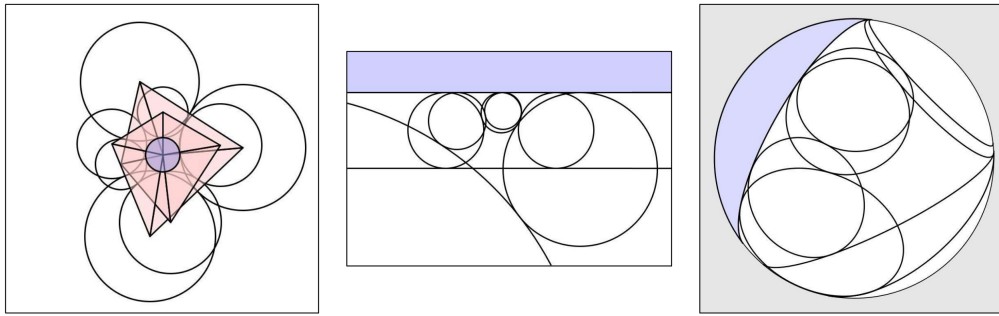

**Figure 10.** Images of a simply branched 8-flower.

Figure 10 has three images of the same branched 8-flower under different Möbius transformations: one Euclidean, one normalized, and one on $\mathbb{P}$. Flowers laid out using schwarzians can easily end up with petals enclosing $\infty$, as on the right; visualization on the sphere is a slippery business.

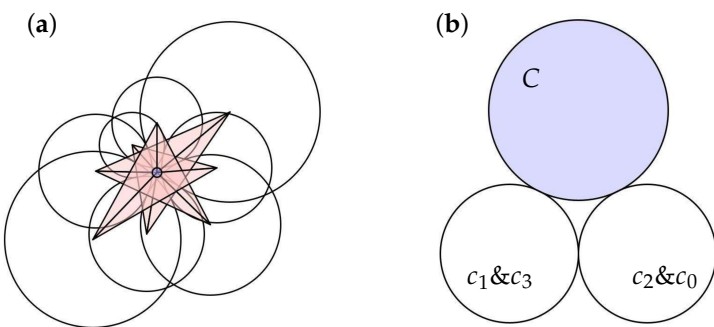

**Figure 11.** Two branched flowers; (**a**) is a typical flower with branching of order 2; (**b**) s branched 4-flower with extraneous tangencies.

Figure 11a illustrates a typical flower with branching of order two, with its eight petals (and eight faces) wrapping three times around $C$. Figure 11b, on the other hand, is a cautionary example: it displays a branched 4-flower. What constitutes a flower depends on context: Are extraneous tangencies allowed? Must the flower be able to live in a larger circle packing? Etc. The standard requirement is that a flower wrapping $k$ times about its center will require $n \geq (2k+1)$ petals. This would hold if we required schwarzians $s$ strictly less than 1 (i.e., $u > 0$). In light of Lemma 3, the 4-flower of Figure 11b is a limit: let $u \downarrow 0$ so $u' = 2/(3u) \uparrow \infty$. The petals $c_1$ and $c_3$ are identical, as are $c_2$ and $c_0$. This is not a situation that would occur in the practice of discrete function theory.

The adjustments in our machinery to accommodate branching are described in Situation 4 and Figure A4. Given $n-3$ parameters, we are still able to compute the remaining three to form a packing label. Indeed, we can still write $u_{n-2} = \mathfrak{U}_n(u_1, \cdots, u_{n-3})$, but we must accept the function $\mathfrak{U}_n(\cdots)$ as representing an **algorithm** rather than an explicit formula.

## 6. Special Classes of Flowers

To illustrate what we have developed, we consider five distinguished families of flowers whose schwarzians can be computed explicitly. Examples are shown in their native environments in Figure 12. Once again, the author would point to the intriguing relations that emerge in these beautiful but elementary cases.

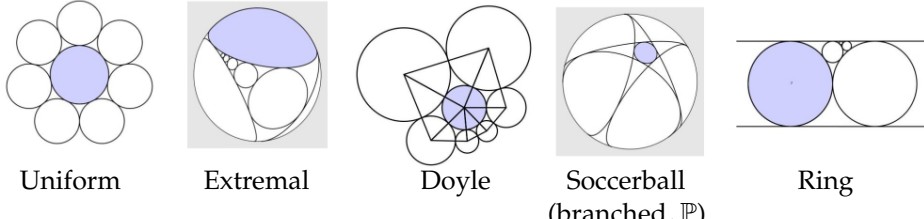

| Uniform | Extremal | Doyle | Soccerball (branched, $\mathbb{P}$) | Ring |

**Figure 12.** Examples of special flower classes.

### 6.1. Uniform Flowers

There is a unique "uniform" univalent $n$-flower for each $n$. The model $n$-flower might be a Euclidean one whose $n$ petals all share the same radius, as in Figure 12. These are a natural, unbiased starting point when first encountering flowers, and deviations from uniformity can be useful (even in computations; see the "uniform neighbor model" of [16]). Figure 13 illustrates several uniform flowers in our normalized form. Symmetry insures that all $n$ schwarzians are identical and one can observe reflective symmetry in the layouts.

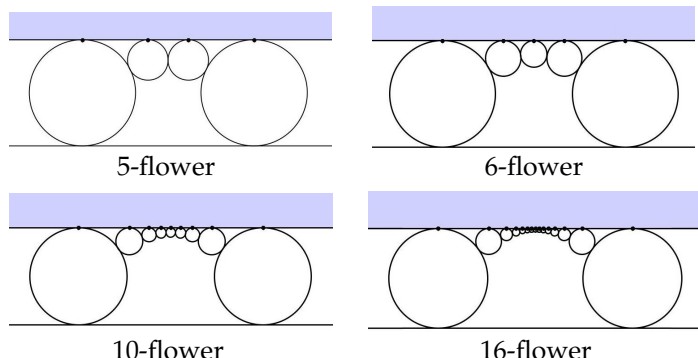

5-flower    6-flower

10-flower    16-flower

**Figure 13.** Samples of "uniform" flowers.

There is some regularity that your eye may pick up in Figure 13. Figure 14 explains that feeling: in every case all, $n$ petals are tangent to a common circle, shaded in the figure. In the model Euclidean setting, this circle is the one circumscribing the flower. (Note, conversely, if such a circle tangent to all the petals exists, then the flower is uniform.)

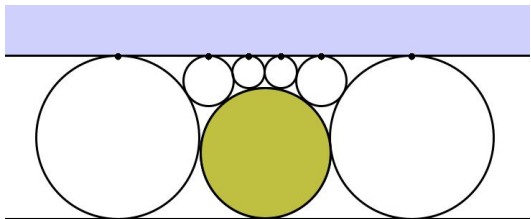

**Figure 14.** The hidden circle.

The key question, is of course "What is the schwarzian for a uniform $n$-flower?". We will label this value as $\mathfrak{s}_n$. Three cases are already known: $\mathfrak{s}_3 = 1 - 1/\sqrt{3}$,

$\mathfrak{s}_4 = 1 - \sqrt{2/3}$ (from Lemma 3), and $\mathfrak{s}_6 = 0$. In Appendix A.2 we establish the closed form $\mathfrak{s}_n = 1 - \frac{2}{\sqrt{3}} \cos(\pi/n)$. Here, is a sampling of values:

$$\mathfrak{s}_3 = 1 - 1/\sqrt{3} \sim 0.422650 \qquad \mathfrak{s}_4 = 1 - \sqrt{2/3} \sim 0.183503$$
$$\mathfrak{s}_5 \sim 0.065828 \qquad \mathfrak{s}_6 = 0$$
$$\mathfrak{s}_9 \sim -0.085064 \qquad \mathfrak{s}_{12} \sim -0.115355$$
$$\mathfrak{s}_{20} \sim -0.140485 \qquad \mathfrak{s}_{50} \sim -0.152422$$

The same computations work for branched uniform flowers, giving the more general formula in (A7), which will be used in Section 6.3.

### 6.2. Extremal Flowers

There is a unique "extremal" univalent $n$-flower for each $n$. Indeed, suppose one of the schwarzians $s$ of a univalent $n$-flower equals the lower bound $1 - (n-2)/\sqrt{3}$. Designating that as the petal $c_0$, (A1) implies that the normalization process can only lead to a flower like that of Figure 15a (for $n = 7$).

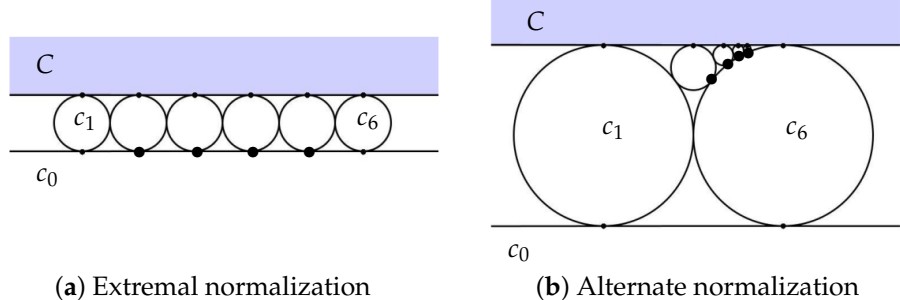

(**a**) Extremal normalization     (**b**) Alternate normalization

**Figure 15.** Normalizations of an "extremal" univalent flower.

As can be seen in this normalization, extraneous tangencies allow $c_1$ and $c_{n-1}$ to serve as the centers of 3-flowers, implying that their schwarzians $s_1, s_{n-1}$ both equal $\mathfrak{s}_3 = 1 - 1/\sqrt{3}$. Next, consider any of the remaining petals $c_j, j = 2, \cdots, n-2$. First, $c_j$ acts as the center of a 3-flower if we include just one of its horizontal neighbors $c_{j+1}$ or $c_{j-1}$ along with the two half planes. As part of this 3-flower, the horizontal edge to the neighbor has schwarzian $1 - 1/\sqrt{3}$. Note at the same time that $c_j$ acts as the center of a 4-flower if you throw in both its horizontal neighbors. Knowing the horizontal schwarzians, we can apply Lemma 3 to compute the schwarzian $1 - 2/\sqrt{3}$ for the vertical edges. In particular, we have the following

**Lemma 4.** *For every $n \geq 3$, there is a unique univalent extremal $n$-flower, and its set of schwarzians is given by*

$$\{\frac{\sqrt{3}-1}{\sqrt{3}}, \frac{\sqrt{3}-2}{\sqrt{3}}, \frac{\sqrt{3}-2}{\sqrt{3}}, \cdots, \frac{\sqrt{3}-2}{\sqrt{3}}, \frac{\sqrt{3}-1}{\sqrt{3}}, \frac{(n-2)}{\sqrt{3}}\}.$$

Figure 15b shows an alternate normalization of the same extremal 7-flower, hence with a shifted list of the same schwarzians.

### 6.3. Soccerball Flowers

We next go into detail about the soccerball circle packings discussed in Section 2.2 and displayed there in Figure 2a. The highly symmetric nature of these packings allows us to calculate their intrinsic schwarzians explicitly—a rare opportunity.

The construction of the soccerball packings is described more fully in [4]. Briefly, the complex $K$ has 42 vertices, 12 of degree 5 and the rest of degree 6. The packing $P_K$ on the left in Figure 2a is the maximal packing for $K$ and is called the soccerball packing because its dual faces form the traditional soccer ball pattern, breaking the sphere into 5- and 6-sided polygonal regions. That on the right, $P$, is a branched packing for $K$, with simple branching at each of the degree 5 vertices. The mapping $F : P_K \to P$ is a key instance of a discrete rational function.

The ubiquitous symmetries within $K$, $P_K$, and $P$ allow one to establish these facts: (a) $K$ has only two types of edges, those with ends of degrees 5 and 6 and those whose ends are both degree 6. (b) In each of $P_K$ and $P$, all circles of degree 5 share one radius, while all of degree 6 share another. (c) These facts imply that in each of $P_K$ and $P$, there are only two intrinsic schwarzians: $s$ for edges of degree 5 flowers and $s'$ between degree 6 circles. (d) And finally, it follows that the degree 5 flowers are uniform while the degree-6 schwarzians take the alternating pattern $\{s, s', s, s', s, s'\}$

Working in $P_K$ first, (d) above and (A6) imply that $u = 1 - s = \frac{2}{\sqrt{3}} \cos(\pi/5)$. It remains to compute $u' = 1 - s'$. However, it turns out we can work more generally. Consider any univalent degree 6 flower whose label has the alternating form $\{u, u', u, u', u, u'\}$. Examples are shown in Figure 16.

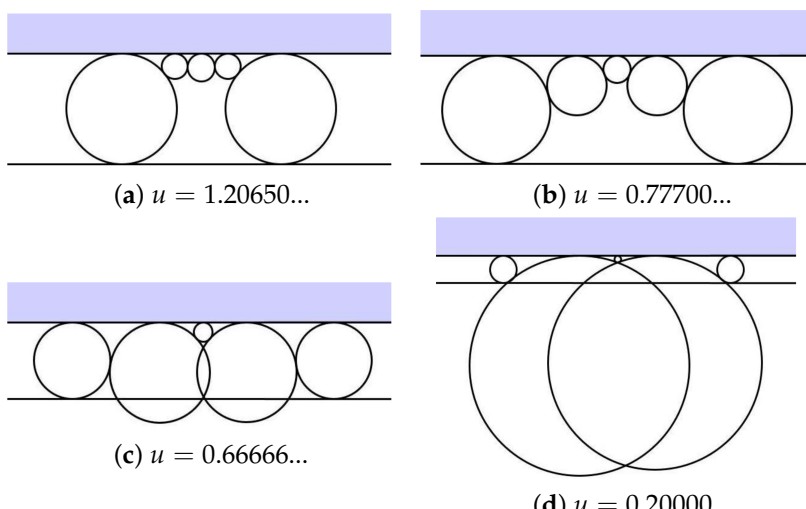

(**a**) $u = 1.20650...$      (**b**) $u = 0.77700...$

(**c**) $u = 0.66666...$

(**d**) $u = 0.20000...$

**Figure 16.** Six-flowers with alternating schwarzians.

Applying $\mathfrak{U}_6$ and some simplification, we have

$$u = \mathfrak{U}_6(u', u, u') \implies u = \frac{u'u}{3u'uu' - u' - u'} \implies uu' = 1. \tag{19}$$

Surprise: $uu' = 1$. In our particular case, we conclude for $P_K$ that

$$s = 1 - \frac{2}{\sqrt{3}} \cos(\pi/5) \sim 0.065828 \tag{20}$$

$$s' = 1 - \frac{\sqrt{3}}{2} \sec(\pi/5) \sim -0.070466.$$

Another feature of these special degree 6 flowers might catch your eye in Figure 16a–c: one can show that, for any $s \leq \mathfrak{s}_3$, a normalized flower with label $\{s, s', s, s', s, s'\}$ will lead to circle $c_4$ having a radius of $r_4 = 1/4$. The flower in Figure 16d shows that this fails for non-univalent cases: when $s > \mathfrak{s}_3$, $c_3$ and $c_5$ overlap. (Incidentally, when $s$ is precisely

equal to $\mathfrak{s}_3$, $c_3$ and $c_5$ are tangent and the value $1/4$ for $c_4$ comes directly from the Descartes Circle Theorem.)

We turn now to the branched packing $P$ on the right in Figure 2a. Each of the 12 degree flowers is branched, so the Riemann–Hurwitz relations imply that $P$ is a seven-sheeted covering of the sphere. The packing is very difficult to interpret: the degree 5 circles are smaller now, but the degree 6 are quite huge—nearly hemispheres. This and the seven sheetedness make individual degree 6 circles very hard to distinguish in Figure 2a, so an isolated 5-flower is shown in Figure 12. Many facts about $P_K$ persist in $P$: as before, there are just two schwarzians, $s, s'$; the degree 5 vertices have uniform flowers; and $s, s'$ alternate in the degree 6 flowers. The normalized layout for one of these 6-flowers is shown in Figure 16d.

We can now compute $u = 1 - s$ using the general expression (A7) from the Appendix A.2. A simple branched 5-flower wraps twice around its central circle; this means for a uniform flower that each face subtends an angle $\theta = 4\pi/5$, implying that $\alpha = \theta/2 = 2\pi/5$. Using (A7) and then (19), we conclude for $P$ in Figure 2a,

$$s = 1 - \frac{2}{\sqrt{3}}\cos(2\pi/5) \sim 0.643178 \tag{21}$$

$$s' = 1 - \frac{\sqrt{3}}{2}\sec(2\pi/5) \sim -1.802517.$$

As a final comment, we observe that, for this very special complex $K$, our analysis extends to other pairs of schwarzians $s, s'$ satisfying (19), meaning such that $(1 - s)(1 - s') = 1$. One obtains a family of projective circle packings $P_s$ which live on coned spheres. I am not prepared to address the range of possible values—an interesting question in itself—but one can choose $s$ to interpolate between (20) and (21) and certainly to extend beyond that range. For each appropriate $s$, the process described in Section 4, starting with an arbitrary mutually tangent triple of circles for some base face and then using the schwarzians to lay out the remaining circles, generates a circle packing $P_s$. Only when $s$ takes its value from (20) or (21) (i.e., $P_s = P_K$ or $P_s = P$, respectively) is $P_s$ a traditional circle packing on the Riemann sphere $\mathbb{P}$. In all other cases, the face-by-face construction produces a topological sphere $\mathbb{P}_s$ with spherical geometry, save for the 12 points associated with degree 5 vertices. These are clearly 12 cone points. The symmetry group of $K$ is the dodecahedral group, so there exists a Möbius transformation $P_s$ making the singularities indistinguishable, that is, they all share a common cone angle $\gamma$. To illustrate, if $s = -0.321284$, then `CirclePack` tells us that $\gamma = 3\pi$. It is left to the curious reader to work out the precise relationship between $s$ and $\gamma$.

Incidentally, imposing symmetry was necessary here since cone angles are subject to change under Möbius transformations. Irrespective of the construction of $P_s$ (i.e., of the initial face $f_0$), the traditional angle sums $\theta_v$ at all vertices of degree 6 will be $2\pi$. However, the angle sums at the degree 5 vertices may no longer share a common value. The only exceptions are our two special cases: when $s$ takes the value in (20) or (21), then the degree 5 vertices have angle sums $\gamma = 2\pi$ or $4\pi$, respectively, regardless of the choices of $f_0$ in the construction. I personally find this curious—this persistence of angle sums when they are multiples of $2\pi$ is nearly a packing criterion.

### 6.4. Doyle Flowers

An early and fascinating chapter in the story of circle packing involves a pattern for hex (degree 6) flowers observed by Peter Doyle. We investigate this two-parameter family of flowers here, but the interested reader can discover the beauty of the "Doyle spirals" that they lead to in [17]. In addition to providing an obvious instance of a discrete exponential

function, these spirals raised the oldest question from the foundational period that remains open, as posed by Peter Doyle: *Do there exist any circle packings of the plane in which every circle has degree 6 other than the familiar "penny packing", in which all circles share a common radius, and the (coherent) Doyle spirals?*

The Doyle flowers involve two radius parameters, $a$ and $b$. If the center $C$ has a radius of 1, then the petal radii take the form

$$\{a, b, b/a, 1/a, 1/b, a/b\}, \quad a, b > 0. \tag{22}$$

Remarkably, regardless of $a$ and $b$, this pattern of radii will always form a 6-flower around $C$. More significant in the search for schwarzians is the fact that the six triangular faces of that flower fall into three similarity classes. For each $j = 1, \cdots, 6$, let $e_j$ be an edge, $f_j$ and $g_j$ be the neighboring faces, and $\mathfrak{p}_j$ the patch formed by their union. For each $j = 1, 2, 3$, one can check that there is a similarity $\Lambda : \mathfrak{p}_j \longrightarrow \mathfrak{p}_{j+3}$ with $\Lambda(e_j) = e_{j+3}$. As a consequence, the sequence of schwarzians takes the form

$$\{s_1, s_2, s_3, s_1, s_2, s_3\}. \tag{23}$$

View this pattern in light of what we know about general 6-flowers: namely that $u_1 = \mathfrak{U}_6(u_1, u_2, u_3)$; solving for $u_3$ gives

$$u_3 = \frac{u_1 + u_2}{3u_1 u_2 - 1}. \tag{24}$$

In other words, using $u_1, u_2$, we have a new 2-parameter representation of the Doyle flowers in the space $\mathfrak{F}_6$.

It would be interesting to find the relationship between parameters $a, b$ and $u_1, u_2$ (or $s_1, s_2$). However, there are more challenging questions that the reader might like to take on. First, sticking with the Doyle setting for a moment, note that the pattern of a single Doyle flower propagates to an infinite spiral, all of whose flowers share the identical set of schwarzians. The combinatorics underlying all Doyle spirals is that of the hexagonal lattice $H$, easily recognized as the planar lattice behind the penny packing. Within $H$ are three families of parallel axes. What the results above show is that, for a given Doyle spiral, all edges within one of these families share the same schwarzian.

The challenge now is to conceive of other conditions on schwarzians analogous to those of (24). What patterns, what families of flowers, might emerge? In addition, are there patterns for flowers that automatically propagate to larger, perhaps infinite, configurations of circles? Examples might contribute to discrete function theory as Doyle spirals have contributed discrete exponential functions.

### 6.5. Ring Lemma Flowers

In circle packing, the well-known and important "Ring Lemma" provides a lower bound $c(n)$ for the ratio $r/R$ of petal radii $r$ to the center's radius $R$ in any univalent Euclidean $n$-flower. First introduced in [5], the extremal situations and sharp constants were obtained in [18] and were shown to be reciprocal integers in [19]. Of course, we are focused on schwarzians not radii, so we work in our normalized setting. Figure 17 suggests how the extremal normalized flowers develop in a nested fashion, with the extremal $n + 1$-flower being obtained from the extremal $n$-flower by adding the largest possible circle to its smallest interstice. Continuing this *ad infinitum*, we arrive at what we might term an $\infty$-flower.

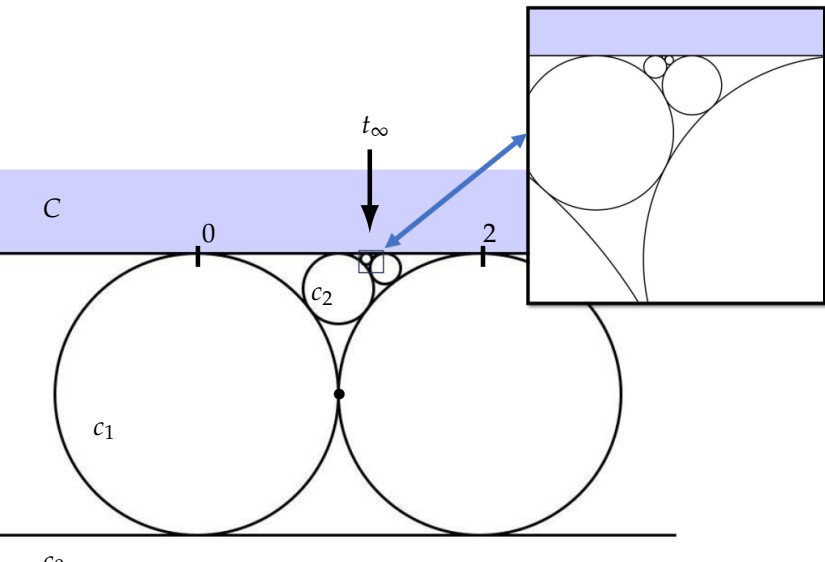

**Figure 17.** Nested "ring" flowers.

At any stage in this development, the current flower is rife with extraneous tangencies. Indeed, at a given stage, we have an $n$-flower whose smallest interstice is formed by $C$ and its two smallest petals. When we plug the new petal into that interstice to form an $n + 1$-flower, the tangency between those two petals becomes extraneous.

The packing's features allow us to compute precise schwarzians. Figure 18 focuses in on the interstice where a new petal, the blue one, is being added. The red and green are the smallest previous petals. Reindexing to accommodate the new petal, we assume that the green circle is $c_{j-1}$, the red is $c_{j+1}$, and the new blue is $c_j$. There are extraneous tangencies, but nonetheless, functionally, $c_{j+1}$ is degree 4, $c_{j-1}$ is degree 5, and of course $c_j$ is degree 3. (In alternating stages, the green would be on the right and the red on the left.)

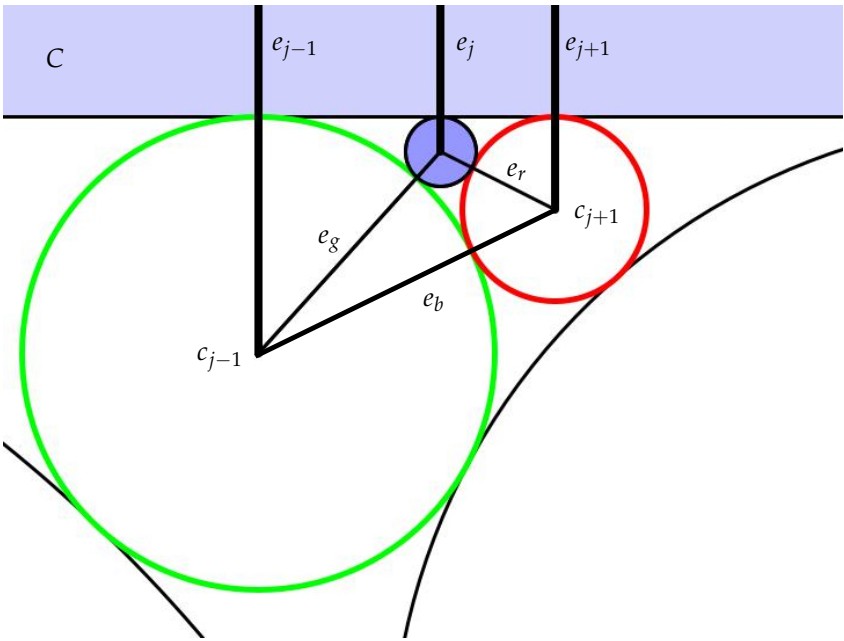

**Figure 18.** Inserting a new petal.

Our local goal is the schwarzians $s_{j-1}, s_j$, and $s_{j+1}$ for the vertical edges $e_{j-1}, e_j$, and $e_{j+1}$, though we need the schwarzians for the edges $e_r, e_g, e_x$ along the way. We will work in the $u$-variables.

The blue petal, $c_j$, is degree 3, so Lemma 2 gives $u_j = u_r = u_g = 1/\sqrt{3}$. The red petal, $c_{j+1}$, is degree 4 and has edges $e_r$ and $e_x$. Lemma 3 implies that $u_r u_x = 2/3$, so knowing that $u_r = 1/\sqrt{3}$, we conclude that $u_x = 2/\sqrt{3}$. Finally, note that the green petal, $c_{j-1}$, has degree 5 and successive edges $u_x, u_g, u_{j-1}$. Knowing $u_x$ and $u_g$ implies $u_{j-1} = \mathfrak{U}_5(u_x, u_g) = \sqrt{3}$. Summarizing for the target schwarzians, we conclude

$$
\begin{aligned}
s_{j-1} &= 1 - \sqrt{3} \sim -0.732051, \\
s_j &= 1 - 1/\sqrt{3} \sim 0.422650, \\
s_{j+1} &= 1 - 2/\sqrt{3} \sim -0.154701.
\end{aligned}
\tag{25}
$$

So, what do we conclude about the schwarzians of a full flower? Observe that, when a new petal is added in our construction, it converts its smaller neighbor, degree 3, in the previous step, to degree 4, while it converts its larger neighbor to degree 5. That larger neighbor, $c_{j-1}$ in Figure 18, remains unchanged thereafter, so its schwarzian remains at $1 - \sqrt{3}$. On the other hand, the schwarzian for the smaller neighbor, $c_{j+1}$, is only temporary, as it will change with the next added petal. So, here is the typical sequence for a Ring Lemma $n$-flower, stated in the alternate $u.$-variables:

$$
\{\sqrt{3}, \cdots\cdots, \sqrt{3}, \frac{1}{\sqrt{3}}, \frac{2}{\sqrt{3}}, \sqrt{3}, \cdots\cdots, \sqrt{3}, \frac{1}{\sqrt{3}}\}.
$$

With every increase in $n$, the $2/\sqrt{3}$ will convert to $\sqrt{3}$, the $1/\sqrt{3}$ will convert to $2/\sqrt{3}$, and a new $1/\sqrt{3}$ will be inserted between them.

Past experience with the Ring Lemma suggests that one should not leave these flowers without looking around for interesting numerical features. In [19] and [20], the Fibonacci sequence, the Descartes Circle Theorem, and the golden ratio all play significant roles. In our normalized setting, we can add Farey numbers to that list.

So, let us look around! As visually suggested in Figure 18, the local picture around a new petal has a static asymptotic limit. We have seen that $u_j = 1 - s_j = 1/\sqrt{3}$, so applying (15) and adjusting the indexing, we see this recurrence relation among the radii:

$$
\frac{1}{\sqrt{r_{j+1}}} = \frac{1}{\sqrt{r_j}} + \frac{1}{\sqrt{r_{j-1}}}.
$$

This is a generalized Fibonacci pattern and is precisely the recurrence observed in ([20], §4). There, one can conclude that

$$
\frac{r}{r'} \longrightarrow (\frac{1+\sqrt{5}}{2})^2 = \tau^2,
$$

where $r'$ is the radius of a new petal, $r$ is the radius of the previous new petal, and $\tau$ is the famous Golden Ratio.

How do Farey numbers enter the picture? Caution: for this discussion, we must scale our normalized Ring Lemma flowers by $1/2$. Thus, the tangency points $t_j$ and radii $r_j$ are now scaled by $1/2$, putting all the tangency points in $[0, 1]$.

One can deduce from the Descartes Circle Theorem that, if a circle is placed in the interstice of three mutually tangent circles whose bends (reciprocal radii in the terminology of F. Soddy [21]) are integers, then that circle's bend will also be an integer. In our construction, we continually put new circles in interstices. One can prove inductively that all radii (after our scaling by $1/2$) are reciprocal integers. From this, one can conclude that all the tangency points $t_j$ are rational numbers. Indeed, these all fall into what is known as the "Farey sequence" in $[0, 1]$ and are subject to the counterintuitive Farey arithmetic. Consider

a tangency $t_j$ for a new circle in our construction, between the tangency points $t$ and $t'$ for the previous two new circles. We may write $t = a/b$ and $t' = a'/b'$ as rational numbers in the lowest terms. From the Descartes Circle Theorem, one can show that

$$t_j = \frac{a + a'}{b + b'}. \tag{26}$$

To see the overall pattern of (rescaled) tangency points, we will redefine the indexing as a sequence $\{t_j\}$. Here, $t_0 = 0$, $t_1 = 1$, and thereafter, let $t_j$ denote the tangency point of the next new petal added, so $t_j$ always falls between $t_{j-1}$ and $t_{j+1}$. (This indexing is not that used for individual $n$-flowers.) Now write $t_0 = 0/1$ and $t_1 = 1/1$ and then repeatedly apply (26). (There is one choice involved; after $t_2 = 1/2$ in Figure 17, we chose $t_3 = \frac{2}{3}$ rather than $t_3 = \frac{1}{3}$.) Here, then, are the first few values

$$\{\frac{0}{1}, \frac{1}{1}, \frac{1}{2}, \frac{2}{3}, \frac{3}{5}, \frac{5}{8}, \frac{8}{13}, \frac{13}{21}, \frac{21}{34} \cdots \cdots \}.$$

As one can see, $t_j = \mathcal{F}_j / \mathcal{F}_{j+1}$, where $\mathcal{F}_j$ is the $j$th Fibonacci number. It is well known that this ratio converges to $1/\tau$. In other words, the new petals in the infinite flower suggested by Figure 17 converge to the point $t_\infty = 2/\tau$. Are not circles grand!

## 7. Conclusions

This paper has introduced a discrete analogue of the classical schwarzian derivative, primarily with the goal of manipulating circle packings in spherical geometry. That goal is not yet within reach, but the first steps have been taken. Using intrinsic schwarzians, this paper has characterized packing edge labels for flowers, which are the basic building blocks of every circle packing. How this fits into the broader enterprise, the task of computing the packing edge labels $S$ for general complexes $K$, remains to be seen. In Section The Difficulty, "criteria" and "monotonicities" were identified as valuable ingredients. Our work with flowers provides cumbersome but easily computable criteria: we can check whether a set $S$ of intrinsic schwarzians for $K$ is a packing edge label. The outlook for monotonicities within schwarzian labels, however, remains quite cloudy. An adjustment of the label for one edge typically affects two flowers. Generating an adjustment that is favorable in moving toward a packing label is the challenge. The author is currently experimenting with maximal packings $\mathcal{P}_K$ for topological spheres $K$, since the correct labels $S$ are already known. Starting with a random label and then repeatedly fixing the labels for randomly chosen flowers, one hopes to see how the adjustment of schwarzians reverberate through the network and how they can be driven toward $S$. What we learn might then be applied to cases with branching as well. It would help if there were some Möbius invariant quantity that one could maximize or minimize. Even then, since the criteria for branching do not involve explicit expressions, classical tools such as differentials may not be available. These difficulties are unfortunate, as branched packings are our primary targets.

In conclusion, although branched packings are handled quite routinely in the disc or the plane, where the controlling parameters are radii and angle sums, the sphere is a tougher environment. Whether there is some way to deploy the properties we have developed for flowers to the computational hurdles with larger packings remains the key open question.

**Funding:** This research received no external funding.

**Data Availability Statement:** No new data were created or analyzed in this study. Data sharing is not applicable to this article.

**Conflicts of Interest:** The author declares no conflicts of interest.

## Appendix A

Here, we detail various computations with intrinsic schwarzians. We start with four situations underlying our construction of normalized flowers from given schwarzians. Next, we compute the schwarzians for uniform flowers. Finally we work out the relationship between a discrete Schwarzian derivative and the intrinsic schwarzians of domain and range.

*Appendix A.1. Layout Computations*

We work with flowers in their normalized positions; see Figure 5 for the notation. Note that $C$ and $c_0$ are tangent at infinity, so the imaginary axis represents the edge $e_0$ between them, associated with schwarzian $s_0$.

In computing the remaining petals, we encounter three situations, and possibly a fourth if there is branching. In each there is an edge $e$ of interest connecting the upper half plane to a petal circle (the shaded one) whose position has already been established. There is also an "initial" neighboring petal (green) which is also in place. Our task is to find data for the companion "target" petal (red), that which is opposite to the initial petal across edge $e$. The shaded face $f$ is that formed by the central circle, the shaded circle, and the initial circle. We are given the initial data for the two petals in place and the schwarzian $s$ for $e$ and show the computations of data for the target circle; in particular, we compute its tangency point $t$ and its radius $r$. The formulas we arrive at are easier to work with if we introduce $u = 1 - s$ as an alternative to the variable $s$ itself. Situations 1-3 are illustrated as they would appear in un-branched flowers. The computations, however, are fully general, as we discuss in Situation 4.

**Situation 1.** We begin with the edge $e = e_0$, connecting the two half planes as illustrated in Figure A1. The petal $c_0$ (a half plane) and the initial petal $c_1$ are in place as part of our normalization. The target is the clockwise neighbor of $c_0$, namely the petal $c_{n-1}$. Being tangent to both half planes, its radius is $r_{n-1} = 1$ and we need only compute its tangency point $t_{n-1}$ from the schwarzian $s_0$.

Let $s = s_0$. The computations involve the Möbius transformation $M_s$ (see (7)) and the Möbius $m_f$ mapping the points $\{1, \omega, \omega^2\}$ to $\{\infty, -2i, 0\}$, and hence mapping $C_v$ to the upper half plane.

$$m_f \circ M_s^{-1} = \begin{bmatrix} 2i & -\sqrt{3} + i \\ -1/2 + \sqrt{3}/2\,i & 1/2 - \sqrt{3}/2\,i \end{bmatrix} \begin{bmatrix} 1-s & s \\ -s & 1+s \end{bmatrix}$$

$$= \begin{bmatrix} \sqrt{3}s + (2 - 3s)\,i & -\sqrt{3}(1+s) + (3s+1)\,i \\ -1/2 + \sqrt{3}/2\,i & 1/2 - \sqrt{3}/2\,i \end{bmatrix}$$

Applying this transformation to $C_b$ gives the normalized petal $c_{n-1}$. In particular, applying it to the tangency point $(5 - \sqrt{3}\,i)/2$ in the base patch $\mathfrak{p}_\Delta$ yields the normalized tangency point $t_{n-1}$, expressed using $u_0 = 1 - s_0$:

$$t_{n-1} = 2\sqrt{3}\,u_0 \qquad \text{and} \qquad r_{n-1} = 1. \tag{A1}$$

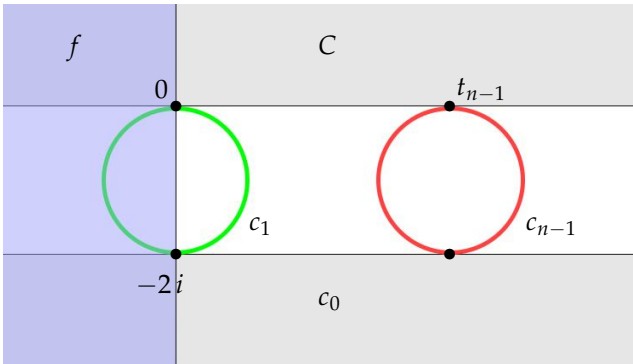

**Figure A1.** Situation 1: layout the red circle.

**Situation 2.** We move now to the edge $e = e_1$ with the target being $c_2$. The relevant schwarzian is $s = s_1$ and the initial petal is the half plane $c_0$, green in Figure A2.

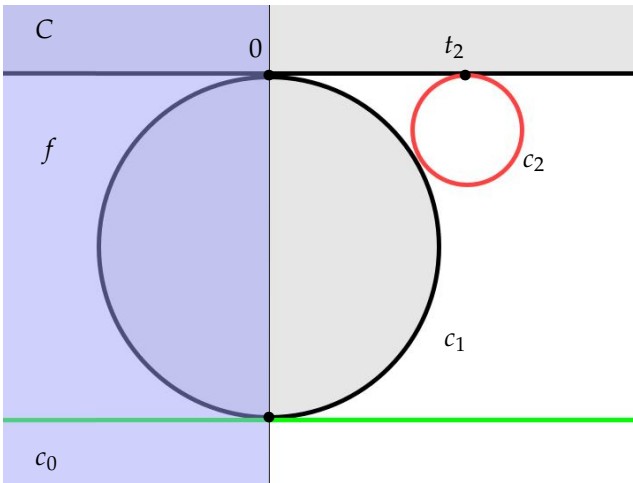

**Figure A2.** Situation 2: layout the red circle.

We proceed by modifying the previous argument. The shaded face $f$ is the same, but the Möbius $m_f$ must now map $\{1, \omega, \omega^2\}$ to $\{0, \infty, -2i\}$. We accomplish this by pre-composing the earlier $m_f$ with a rotation by $\omega^2$. The result is

$$
m_f \circ M_s^{-1} = \begin{bmatrix} \sqrt{3} - i & -\sqrt{3} + i \\ 1 & \dfrac{1}{2} - \dfrac{\sqrt{3}}{2}i \end{bmatrix} \begin{bmatrix} 1 - s & s \\ -s & 1 + s \end{bmatrix}
$$

$$
= \begin{bmatrix} \sqrt{3} - i & -\sqrt{3} + i \\ 1 - 3s/2 + (\sqrt{3}s/2)i & (1 + 3s)/2 - (\sqrt{3}(1 + s)/2)i \end{bmatrix}
$$

Applying this transformation to $C_b$ gives the normalized petal $c_1$. Note that $m_f$ now maps $C_w$ to the upper half plane, so applying the above Möbius to the tangency point $(5 + \sqrt{3}i)/2$ in the base patch yields the displacement to the normalized tangency point $t_2$; simple geometric computations give the radius. We use the variable $u_1 = 1 - s_1$.

$$
t_2 = 2/(\sqrt{3}\,u_1) \qquad \text{and} \qquad r_2 = (t_2)^2/4 = 1/(\sqrt{3}\,u_1)^2. \tag{A2}
$$

**Situation 3.** We are left to treat the generic situation suggested by Figure A3. The edge $e$ goes from the central circle $C$ to the shaded circle, with its schwarzian $s$ and variable

$u = 1 - s$. (Note that the half plane for $c_0$ is no longer necessarily involved.) We assume that the shaded circle has a radius $R$, while the initial green circle has radius $r$. It is convenient to position the shaded circle tangent to $C$ at the origin, and then our goal is to compute the tangency point $\delta$ (the displacement from 0) and the radius $\rho$ of the red target circle.

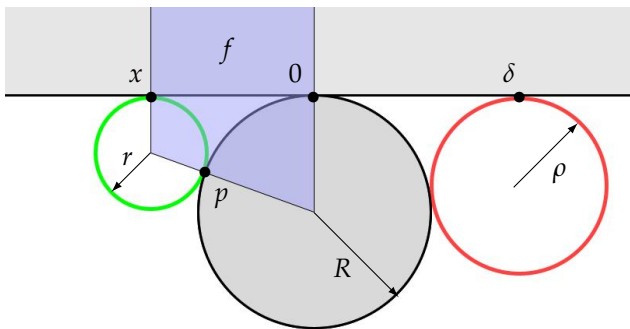

**Figure A3.** Situation 3: layout the red circle.

Elementary geometric computations yield

$$x = -2\sqrt{rR}, \qquad p = -\left(\frac{2R}{R+r}\right)(\sqrt{rR} + (r+R)\,i).$$

The following Möbius transformation $m$ will convert this generic situation to Situation 2. Namely, $m$ maps $\{x, p, 0\}$ to $\{\infty, -2i, 0\}$, so the configuration of Figure A3 morphs into that of Figure A2.

$$m = \begin{bmatrix} 1 + \sqrt{r/R}\ i & 0 \\ (\sqrt{R/r} + i)/2 & R + \sqrt{rR}\ i \end{bmatrix}$$

The tangency point $t_2$ in Figure A2 corresponds to the tangency point $\delta$ in Figure A3, so $\delta$ is obtained by applying $m^{-1}$ to $t_2$. An annoying calculation gives, in the alternate variable $u$,

$$\delta(u, r, R) = \frac{2R}{(\sqrt{3}\,u - \sqrt{R/r}\,)} \quad \text{and} \quad \rho = \frac{1}{(\sqrt{3}\,u/\sqrt{R} - 1/\sqrt{r})^2}. \tag{A3}$$

We will also need to reverse these computations in a particular situation in order to compute $s$. The situation is this: the values $r$ and $R$ are known, $\delta$ is positive, and the computed radius $\rho$ comes out to be 1. What is $s$? We compute $u$, then $s = 1 - u$.

$$\text{When } R, r \text{ are known, } \delta > 0, \text{ and } \rho = 1: \qquad u = \frac{\sqrt{R} + \sqrt{R/r}}{\sqrt{3}}. \tag{A4}$$

Situation 2 is the limiting case of Situation 3 when $r$ grows to $\infty$, so (A2) follows from (A3). Also, note that, when applying (A3), the quantity $\delta$, which represents the displacement of the target circle from its shaded neighbor, can be zero or negative. An example is the branched flower of Figure 6c: with initial circle $c_2$, the displacement of the target $c_4$ from $c_3$ is negative. This puts us in the following branching situation.

**Situation 4.** Branching is initiated during a layout step if and only if (A3) results in a displacement $\delta \leq 0$. Figure A4a illustrates the most typical case, with $\delta_j = (t_{j+1} - t_j) < 0$. However, it is laying out the next circle that we need to concentrate on, as shown in Figure A4b. (The color codings are as before; known green and shaded petals in place, a red target petal to be positioned based on the schwarzian of the edge to the shaded circle.)

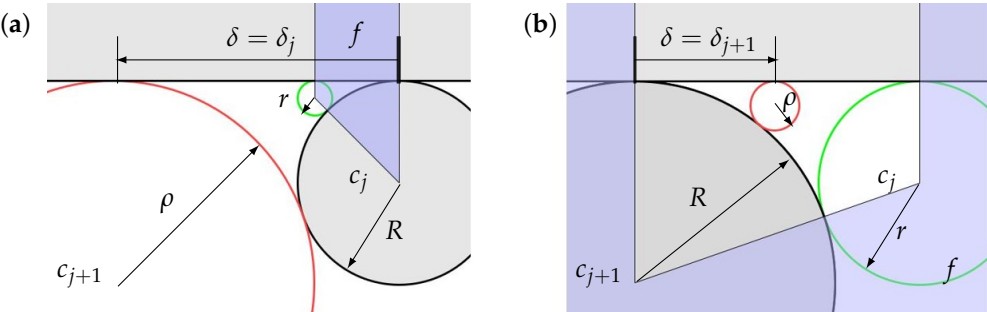

**Figure A4.** Situation 4: (**a**) shows the layout before adding the red circle of (**b**).

By the formula in (A3), when $\delta < 0$, then $(\sqrt{3}u/\sqrt{R} - 1/\sqrt{r}) < 0$. This means, in turn, that our previous expression $1/\sqrt{\rho} = (\sqrt{3}u/\sqrt{R} - 1)$ is no longer true, as it requires absolute values on the right-hand side. Subsequent formulas like those in (10) and (15) fail, and ultimately, $\mathfrak{U}_n$ is no longer represented in a closed formula. This is what makes branched flowers more difficult to manipulate.

Figure A4b is typical of what we refer to as Situation 4. Notice that the new displacement, $\delta_{j+1}$, is again in the positive direction. The computations require a modification of (A3).

When the **previous** displacement was negative, then (A3) becomes

$$\delta(u, r, R) = \frac{2R}{(\sqrt{3}\,u + \sqrt{R/r}\,)} \quad \text{and} \quad \rho = \frac{1}{(\sqrt{3}\,u/\sqrt{R} + 1/\sqrt{r})^2}. \tag{A5}$$

The "previous" step refers to that where $R$ was computed. *A propros* of our earlier comments, the modification here is simply replacing $\sqrt{R}$ by $-\sqrt{R}$ in (A3). (There is one other detail: the standing assumption $u \geq 0$ is also required to ensure that this new displacement $\delta$ is positive).

Another possibility leading to branching is pictured in Figure A5. Namely, when $(\sqrt{3}\,u - 1/\sqrt{R/r}) = 0$ in (A3), so $\delta$ is undefined. In essence, $\delta = \infty$, $R = \infty$, and the petal $c_{j+1}$ is a half plane (i.e., tangent to $C$ at $\infty$). Figure A5 illustrates the situation when placing the next petal $c_{j+2}$, which necessarily has the same radius $r_j$ as $c_j$. For its tangency point, note that Figure A5 is a version of Figure A1. Applying (A1), scaling by $r_j$, and taking the order $t_j, \infty, t_{j+2}$ of the tangencies about $C$ into account, we have $t_{j+2} - t_j = -2\sqrt{3}u_{j+1}r_j$.

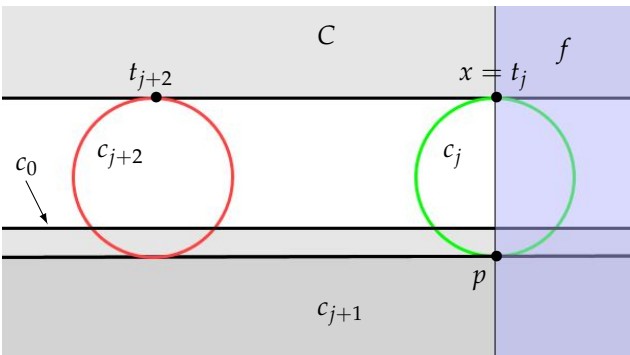

**Figure A5.** Laying out the red circle when $c_j$ is a half plane.

We conclude this subsection by explaining the two exceptions to successful layout as listed in Theorem 1. The exceptional situations occur when $j + 1 = n - 2$ in Figures A4 and A5. Regarding exception (a), if Figure A5 occurs (so $c_{j+1}$ is the penultimate petal $c_{n-2}$), then the Layout Process fails because placing $c_{j+2}$ (i.e., $c_{n-1}$) with mandated

radius 1 is either impossible (if $r_j \neq 1$) or ambiguous (since $u_{n-2}$ is unknown). Regarding exception (b), look to Figure A4b. Though $c_1$ is not pictured here, if the tangency point of the red circle, $t_{n-1}$, is negative (to the left of $t_1 = 0$), then (A1) implies $u_0$ is negative, that is $s_0 > 1$, which is not allowed.

*Appendix A.2. Uniform Petals*

The schwarzians for a uniform $n$-flower take a constant value that we have labeled $\mathfrak{s}_n$. Here, we show that

$$\mathfrak{s}_n = 1 - \frac{2\cos(\pi/n)}{\sqrt{3}}, \quad n \geq 3. \tag{A6}$$

We will base our computations on Figure A6, with $C$ being the unit circle and successive petals $c_{n-1}, c_0, c_1$ sharing a common radius. Our interest is in the schwarzian $s$ for the edge from $C$ to $c_0$.

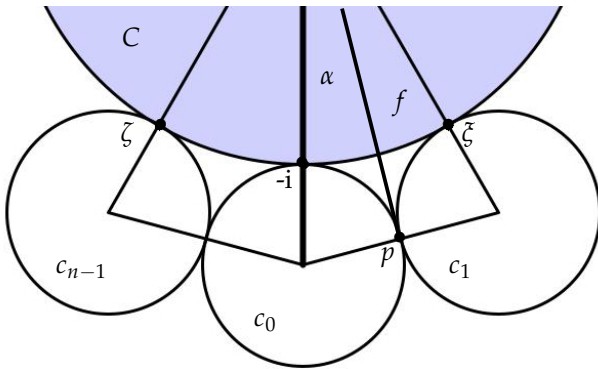

**Figure A6.** Uniform petals: compute the schwarzian.

Let the angle $\alpha$ be one half of the angle at the origin in face $f$ formed by the triple $\{C, c_0, c_1\}$. If these petals were taken from a uniform $n$-flower, then $\alpha = \pi/n$. However, the following computation works for any $\alpha, 0 < \alpha < \pi/2$. Note that the tangency points of the circles are

$$\xi = \sin(2\alpha) - i\cos(2\alpha) \quad \text{and} \quad \zeta = -\sin(2\alpha) - i\cos(2\alpha).$$

Let $T$ denote the Möbius transformation which maps $\{-i, p, \xi\}$ to $\{\infty, -2i, 0\}$, where $p$ is the tangency point between $c_0$ and $c_1$. This transformation puts the four circles in the standard normalized positions as they appear in Figure A1. In particular, $T(\zeta)$ is the tangency point labeled $t_{n-1}$ there. Applying (A1), we conclude that $1 - s = T(\zeta)/(2\sqrt{3})$. I will leave the computation of $T$ to the curious reader, but here is the general result:

$$s = 1 - \frac{2\cos(\alpha)}{\sqrt{3}}, \quad 0 < \alpha < \pi/2. \tag{A7}$$

*Appendix A.3. Special Computations*

We outline two computations referred to in Section 3. These are similar in nature and, though elementary with a symbolic math package, provide great fun via pencil-and-paper. Both involve the restriction of a discrete mapping $F$ between circle packings to a domain patch $\mathfrak{p}$ and its image patch $\mathfrak{p}' = F(\mathfrak{p})$. The related objects involved are the edges $e, e'$, their intrinsic schwarzians $s, s'$, their tangency points $t, t'$, the face mappings $m_f : f \longrightarrow f'$,

$m_g : g \longrightarrow g'$, and the discrete Schwarzian derivative $\sigma = \Sigma_F(e)$. Both situations also involve a Möbius transformation $m(z) = (az + b)/(cz + d)$; we write $m$ in matrix form

$$
m = \begin{bmatrix} a & b \\ c & d \end{bmatrix}, \quad \text{with } ad - bc = 1.
$$

The first computation relates the Schwarzian derivative and the two intrinsic schwarzians $s, s'$. The Schwarzian derivative $\sigma$ arises in

$$
m_g^{-1} \circ m_f = \mathbb{I} + \sigma \begin{bmatrix} t & -t^2 \\ 1 & -t \end{bmatrix}.
$$

For the intrinsic schwarzians, we need to identify these additional Möbius transformations identifying faces:

$$
\begin{aligned}
\text{For } \mathfrak{p} = \{f \,|\, g\}: &\quad \mu_f : f_\Delta \longrightarrow f; \quad \mu_g : g_\Delta \longrightarrow g. \\
\text{For } \mathfrak{p}' = \{f' \,|\, g'\}: &\quad \nu_f : f_\Delta \longrightarrow f'; \quad \nu_g : g_\Delta \longrightarrow g'.
\end{aligned}
$$

Manipulating the expression for schwarzians and taking $m = \mu_f$, we get

$$
\nu_g^{-1} \circ \nu_f = \mu_g^{-1} \circ (m_g^{-1} \circ m_f) \circ m \quad \text{and}
$$

$$
\mu_g^{-1} = \begin{bmatrix} 1 + s & -s \\ s & 1 - s \end{bmatrix} \begin{bmatrix} d & -b \\ -c & a \end{bmatrix}.
$$

Putting these into matrix form gives

$$
\begin{bmatrix} 1 + s' & -s' \\ s' & 1 - s' \end{bmatrix} = \begin{bmatrix} 1 + s & -s \\ s & 1 - s \end{bmatrix} \begin{bmatrix} d & -b \\ -c & a \end{bmatrix} \begin{bmatrix} 1 + \sigma t & -\sigma t^2 \\ \sigma & 1 - \sigma t \end{bmatrix} \begin{bmatrix} a & b \\ c & d \end{bmatrix}.
$$

The many pleasant surprises in a pencil-and-paper simplification yield

$$
\begin{bmatrix} 1 + s' & -s' \\ s' & 1 - s' \end{bmatrix} = \mathbb{I} + \begin{bmatrix} s + \sigma/(c + d)^2 & -(s + \sigma/(c + d)^2) \\ s + \sigma/(c + d)^2 & -(s + \sigma/(c + d)^2) \end{bmatrix},
$$

implying $s' = s + \sigma/(c + d)^2$. Moreover, the expression on the right is associated with the map of $\mathfrak{p}_\Delta \longrightarrow \mathfrak{p}'$ and with the tangency point $\tau = 1$ in its domain. The Schwarzian derivative $s' = s + \sigma/(c + d)^2$ may therefore be rewritten

$$
s' = s + \Sigma_F(e) \cdot m'(1). \tag{A8}
$$

(As a side note, $\Sigma_F(e) \cdot m'(1)$ is real.)

Schwarzian derivatives—both classical and discrete—are unchanged under post-composition by Möbius transformations. Our second computation derives the chain rule for discrete Schwarzian derivatives under pre-composition. We will rely on the notations above, except that $m$ now respresents an arbitrary Möbius transformation and the base patch $\mathfrak{p}_\Delta$ is replaced by the patch $\mathfrak{p}'' = m^{-1}(\mathfrak{p}) = \{f'' \,|\, g''\}$ with its tangency point denoted $\tau$.

We start with the function $F : \mathfrak{p} \longrightarrow \mathfrak{p}'$. Its Schwarzian derivative $\sigma = \Sigma_F(e)$ is derived from the expression

$$
m_g^{-1} \circ m_f = \mathbb{I} + \sigma \begin{bmatrix} t & -t^2 \\ 1 & -t \end{bmatrix} = \begin{bmatrix} 1 + \sigma t & 1 - \sigma t^2 \\ \sigma & 1 - \sigma t \end{bmatrix}.
$$

The issue is, given $m$, what is the Schwarzian derivative for $F \circ m : \mathfrak{p}'' \longrightarrow \mathfrak{p}'$, denoted by $\Sigma_{F \circ m}(e'')$? This is derived from $\nu_g^{-1} \circ \nu_f$, involving the face maps $\nu_f : f'' \longrightarrow f'$ and $\nu_g : g'' \longrightarrow g'$. Note that $\nu_f = m_f \circ m, \nu_g = m_g \circ m$. Therefore,

$$\nu_g^{-1} \circ \nu_f = (m_g \circ m)^{-1} \circ m_f \circ m = m^{-1} \circ (m_g^{-1} \circ m_f) \circ m.$$

Manipulating this, we arrive at

$$\nu_g^{-1} \circ \nu_f = m^{-1} \cdot \left[ \mathbb{I} + \sigma \begin{bmatrix} t & -t^2 \\ 1 & -t \end{bmatrix} \right] \cdot m$$

$$= \mathbb{I} + \sigma \begin{bmatrix} d & -b \\ -c & a \end{bmatrix} \begin{bmatrix} t & -t^2 \\ 1 & -t \end{bmatrix} \begin{bmatrix} a & b \\ c & d \end{bmatrix}.$$

Since $m$ identifies $e''$ with $e$, we have $m(\tau) = t$. Using this to replace $t$ and enjoying further pencil-and-paper work, one arrives at

$$\nu_g^{-1} \circ \nu_f = \mathbb{I} + \frac{\sigma}{(c\tau + d)^2} \begin{bmatrix} \tau & -\tau^2 \\ 1 & -\tau \end{bmatrix}. \tag{A9}$$

This gives our discrete chain rule, which is placed here beside the classical version:

$$\Sigma_{F \circ m}(e'') = \sigma / (c\tau + d)^2 = \Sigma_F(m(e'')) \cdot m'(\tau) \tag{A10}$$
$$S_{\phi \circ m}(z) = S_\phi(m(z)) / (cz + d)^4 = S_\phi(m(z)) \cdot (m'(z))^2.$$

These diverge in that the discrete version involves $m'$ rather than $(m')^2$. The author has no concrete explanation for this difference. It is perhaps worth noting, however, that for mappings between circle packings, the ratios of image radii to domain radii serve as a proxy for the absolute value of the classical derivative; see, for example, [22]. In some sense, these mappings already incorporate a derivative, and this may subtly influence this chain rule.

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
