# Peer review of "A Discrete Schwarzian Derivative via Circle Packing"

_3042-402X_

Round 1
Reviewer 1 Report
Comments and Suggestions for Authors
Reviewer report on the paper “A discrete Schwarzian derivative via circle packing” by K. Stephenson, submitted to Geometry.
In the paper, the author introduces a new tool in the theory of the discrete analytic functions related with the circle packing, viz. Schwarzian derivatives, and closely related with them “intrinsic schwarzians”. The real utility of these concepts and their applicability range still remain to be demonstrated, but the first examples given in the Manuscript look interesting. The Manuscript contains informative and detailed Appendix, which enables to understand the most essential details of the calculations. Also a broad spectrum of applications of these newly introduced tools are given in numerous examples.
The present reviewer believes that the paper does not contain serious flaws, might be interesting to the readers, and recommend acceptance. My only recommendation is to give somewhat more general discussion of the discrete analytic functions, that which is not related only with the spherical packing.
Further, the references to Figures should be consistent: in the Section 1, the author regularly speaks about “Figure 1.2” while there is only Figure 2. Similarly, what is the (1.2) entry of the Möbius transformation mentioned on p. 12? What are “clunky but serviceable Theorems ??” on p. 16?
Reviewer 2 Report
Comments and Suggestions for Authors
The abstract briefly summarizes the motivation, main results, and scope of the paper. The development of a discrete Schwarzian derivative within the framework of circle packing is clearly stated. The results presented appear to be mathematically valid and correct.
The manuscript does not contain a formal Introduction section. This section is essential to provide background context, a literature overview, and the foundational basis for the work.
Add Section 1 as Introduction and clearly mention the objectives of this work, along with its novelty and motivation.
Check Reference 12 and format it according to the journal's guidelines.
Add a conclusion that summarizes the contributions and discusses future work.
There is a typographical error in line 752.
Add application section for this work.
Can you elaborate on how this discrete Schwarzian may lead to new algorithms in spherical geometry?
How does your definition of intrinsic Schwarzian relate to classical conformal invariants?
Is the discrete Schwarzian invariant under Möbius transformations? Please clarify this with a brief proof or remark.
Reviewer 3 Report
Comments and Suggestions for Authors
In this paper the author introduces a faithful discrete analogue of the classical Schwarzian derivative to this theory and develops its basic properties. A companion
localized notion called an intrinsic schwarzian is also investigated. The main concrete
results of the paper are limited to circle packing flowers. The paper
closes with the study of special classes of flowers that occur in the circle packing literature.
The article is interesting for readers, therefore I recommend it for publication.
As a suggestion for the author, the bibliography should be double-checked and standardized.
Round 2
Reviewer 2 Report
Comments and Suggestions for Authors
Accept in present form.